# The Gravitational Process Path (GPP) model (v1.0) – a GIS-based simulation framework for gravitational processes

Volker Wichmann[1,2]

[1]alpS, Centre for Climate Change Adaptation, 6020 Innsbruck, Austria
[2]Laserdata GmbH, 6020 Innsbruck, Austria

*Correspondence to:* Volker Wichmann (wichmann@alps-gmbh.com)

**Abstract.** The Gravitational Process Path (GPP) model can be used to simulate the process path and run-out area of gravitational processes based on a digital terrain model (DTM). The conceptual model combines several components (process path, run-out length, sink filling and material deposition) to simulate the movement of a mass point from an initiation site to the deposition area. For each component several modeling approaches are provided, which makes the tool configurable for different processes like rockfall, debris flows or snow avalanches. The tool can be applied to regional scale studies like natural hazard susceptibility mapping but also contains components for scenario based modeling of single events. Both the modeling approaches and precursor implementations of the tool have proven their applicability in numerous studies, including also geomorphological research questions like the delineation of sediment cascades or the study of process connectivity. This is the first open source implementation, completely re-written, extended and improved in many ways. The tool has been committed to the main repository of the System for Automated Geoscientific Analyses (SAGA) and thus will be available with every SAGA release.

## 1 Introduction

Rapid mass movements like rockfall, debris flows or snow avalanches, are common features in mountainous regions. Due to population growth and the advancing construction of infrastructure and buildings in such areas, rapid mass movements more and more pose a risk to society and can result in severe damages or even disasters. Besides early warning systems and protection measures for disaster prevention, hazard susceptibility zoning, which identifies potentially endangered areas, is required for risk analysis and the creation of hazard maps (Carrara et al., 1991; Fell et al., 2008; Hu et al., 2016).

While physically based dynamic models can be used for detailed analyses of single events (Takahashi et al., 1992; Iverson, 1997; Pudasaini and Hutter, 2007), regional susceptibility mapping needs modeling approaches with minimal data requirements (Aleotti and Chowdhury, 1999; van Westen and Soeters, 2006; Horton et al., 2013). The input parameters of physically based models are often uncertain, which is why simplified conceptual models are used to estimate potentially endangered areas in regional studies (Mergili et al., 2015). An important part of hazard susceptibility zoning is the description of process paths and run-out distances to determine the objects at risk. This requires to know about potential release areas in order to use these as start points in process path models. Potential process initiation sites can be derived by various methods, including

geomorphological field mapping, the combination of index maps, statistical analyses, deterministic approaches (e.g., factor of safety), probabilistic approaches, or neural networks (Aleotti and Chowdhury, 1999). Originating from the derived starting zones, material, or rather mass points, can be routed over a DTM (digital terrain model). This can be done by single or multiple flow direction algorithms, the latter being able to describe lateral spreading away from the slope line (e.g., O'Callaghan and

Mark, 1984; Freeman, 1991; Horton et al., 2013). In order to determine the run-out length, often simple break criteria are used like threshold angles based on horizontal and vertical distances (Lied and Bakkehøi, 1980; Hungr and Evans, 1988; Dorren, 2003; Zimmermann et al., 1997). Other approaches, often based on the mass flow model of Voellmy (1955), are using simplified physically based models considering only the centre of mass but not its deformation (Körner, 1976; Perla et al., 1980; Hegg, 1996; Gamma, 2000; Wichmann and Becht, 2005; Horton et al., 2013).

This paper introduces the Gravitational Process Path (GPP) model version 1.0, an attempt to provide a GIS-based modeling framework for the simulation of process path and run-out area of gravitational processes. The GPP model is a conceptual model, concatenating components for process path determination, run-out calculation, sink filling and material deposition. For each of these components, several well established modeling approaches are implemented and can be chosen by the user. This makes the GPP model configurable for different processes like rockfall, debris flows or avalanches.

Basically, the GPP model simulates the movement of a mass point over a raster DTM from an initiation site to the deposition area. Therefore it includes empirical, stochastic and physically based modeling approaches and provides the option of terrain modification by material deposition during operation. Although some of the implemented approaches are based on simplifying concepts, realistic results can be achieved with the great advantage of requiring only a few input parameters. This makes it possible to use the tool for regional scale studies, but it also includes some components for scenario modeling of single events.

The approaches implemented in the model components have been successfully used for hazard susceptibility mapping (e.g., Zimmermann et al., 1997; Heinimann et al., 1998; Wichmann and Becht, 2004; Mergili et al., 2015; Proske and Bauer, 2016) and geomorphological process studies, e.g. on sediment cascades or process connectivity (e.g., Wichmann et al., 2009; Haas et al., 2012a; Heckmann and Schwanghart, 2013; Heckmann et al., 2016).

For process path modeling, the GPP model includes the single flow direction path finding approach of O'Callaghan and
Mark (1984), also known as the D8 flow direction approach (Jenson and Domingue, 1988), which has been used in various hydrological and geomorphological applications. Besides, a random walk approach as introduced in the dfwalk model by Gamma (1996, 2000) is implemented. It is especially suited for process path delineation of gravitational processes and has been used by various authors for rockfall modeling (e.g., Wichmann and Becht, 2006; Haas et al., 2012b; Proske and Bauer, 2016), debris flow modeling (e.g., Zimmermann et al., 1997; Heinimann et al., 1998; Wichmann, 2006; Mergili et al., 2015)
and avalanche modeling (e.g., Heckmann, 2006; Schmidtner, 2012).

For run-out distance calculation, the GPP model includes several approaches based on the energy line principle (e.g., Heim, 1932; Hungr and Evans, 1988), which have been applied to various processes including rockfall (e.g., Heinimann et al., 1998; Dorren, 2003), debris flows (e.g., Zimmermann et al., 1997) and avalanches (e.g., Körner, 1980). Besides, the 1-parameter friction model of Scheidegger (1975) is implemented, which has been used for rockfall run-out calculations in several studies
(e.g., van Dijke and van Westen, 1990; Meißl, 1998; Dorren and Seijmonsbergen, 2003; Wichmann and Becht, 2005; Haas

et al., 2012b). Finally, the run-out model of Perla et al. (1980), often referred to as PCM model, is included. The PCM model has been applied for avalanche run-out modeling by e.g., Körner (1976), Hegg (1996) and Heckmann (2006). It has also been applied to model debris flows (Rickenmann, 1990; Zimmermann et al., 1997; Heinimann et al., 1998; Gamma, 2000; Wichmann, 2006; Mergili et al., 2012; Mergili et al., 2015) and large rock slides (e.g., Körner, 1976).

The GPP model is the first open source implementation based on previous work of the author, but it is completely reworked and enhanced in various aspects. It is implemented as a tool for the System for Automated Geoscientific Analyses (SAGA, Conrad et al., 2015) and is released as free open-source software (licensed under the GPL). The source code has been committed to the main repository of SAGA hosted at sourceforge.net (https://sourceforge.net/projects/saga-gis/), and binaries are available with every SAGA release.

The paper is structured as follows: Sect. 2 provides an overview of the framework and the model components (process path, run-out, sink filling and deposition). The individual modeling approaches implemented for each component are described in detail in Sect. 3. In Sect. 4 model configurations and application examples for rockfall, debris flow, avalanche and scenario modeling are presented. Finally a discussion and conclusion is provided.

## 2   General model structure

The GPP model is intended to provide a software framework for gravitational process path modeling. It integrates components for process path determination, run-out calculation, sink filling and material deposition. For each of these components, several modeling approaches are implemented. This makes it possible to concatenate modeling approaches as required to simulate the behavior of a certain geomorphological process or to use suitable approaches with regard to the available input data.

Generally, the GPP model routes a mass point, here called particle (following the nomenclature of physics engines), from
an initiation site over a raster DTM to the deposition area. In the GPP model, these initiation sites are organized in so called release areas, made up of one or more grid cells labeled as starting zones in an input raster data set. Such a raster data set has to be derived beforehand, usually by some kind of susceptibility modeling or (field) mapping.

The GPP model computes several model realizations for each start cell (Monte Carlo simulation). The number of model iterations is defined by the user (default: 1000 iterations). The overlay of the model results from all iterations shows the final
model result, i.e. the complete process area (and not individual process paths), as every iteration will show a different result because of the stochastic components in the model.

Besides the components for process path and run-out calculation, the GPP model integrates components, which can modify the DTM in each model iteration by material deposition: there is a model component, which handles natural or artificial sinks, and a component to deposit material on process stop or along the process path. This allows the model to overcome sinks or
to simulate the blocking of a channel by wood and debris. In order to use these components, the GPP model requires an input data set with material heights per start cell.

Fig. 1 shows a basic setup, usually used for gravitational process path modeling on a regional scale. As this setup does not include the filling of sinks, a hydrologically sound DTM must be used. In each model iteration, a particle is initialized using

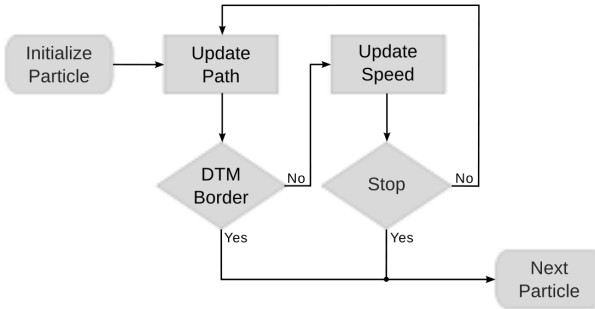

**Figure 1.** Flowchart of a basic GPP model configuration for modeling on a regional scale.

information from its start cell. In a first step, one of the process path models is used to update the particle's path. In case there is no valid process path cell, i.e. the path has reached the border of the DTM or a NoData cell, the particle is deleted and the next particle is initialized. If the next cell in the process path can be determined, one of the run-out models is used to update the speed of the particle, or, in case of an approach based on the energy line principle, the respective angle criterion is checked.

5  In case the particle has stopped, the next particle is initialized. Otherwise, the next cell of the process path is determined.

A model configuration including the filling of sinks is depicted in Fig. 2. This setup requires additional information on the material available per start cell. In case the process path has ended up in a sink, the amount of material available for the particle is checked. This amount of material is then used to fill up the process path upslope while preserving a downward slope, allowing the next particle to overcome the sink. In case the material available in an iteration is not enough or the sink is larger, several model iterations might be necessary to completely fill up the sink. After the attempt to fill the sink, the next particle is initialized.

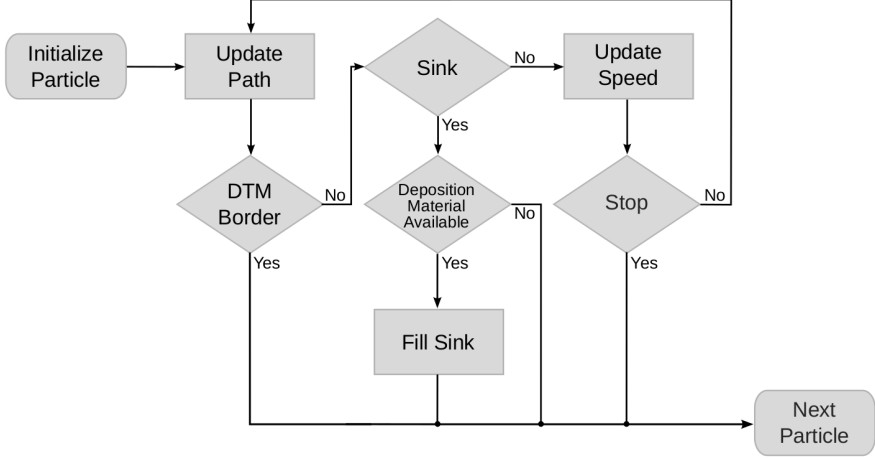

**Figure 2.** Flowchart of a GPP model configuration making use of the sink filling approach.

Fig. 3 shows a fully featured setup of the GPP model, which is usually used for scenario modeling of a single (or some few)
5   events. In this setup, material may be deposited when a particle stops, depending on the chosen deposition model and whether
there is (still) material available for the particle. Then the next particle is initialized. In case the particle has not stopped, it
depends again on the chosen deposition model and the available material whether material is deposited along the process path
or not. Then the next cell of the process path is determined. The deposition of material on stop or based on slope and velocity
along the process path alters the terrain between successive model iterations.

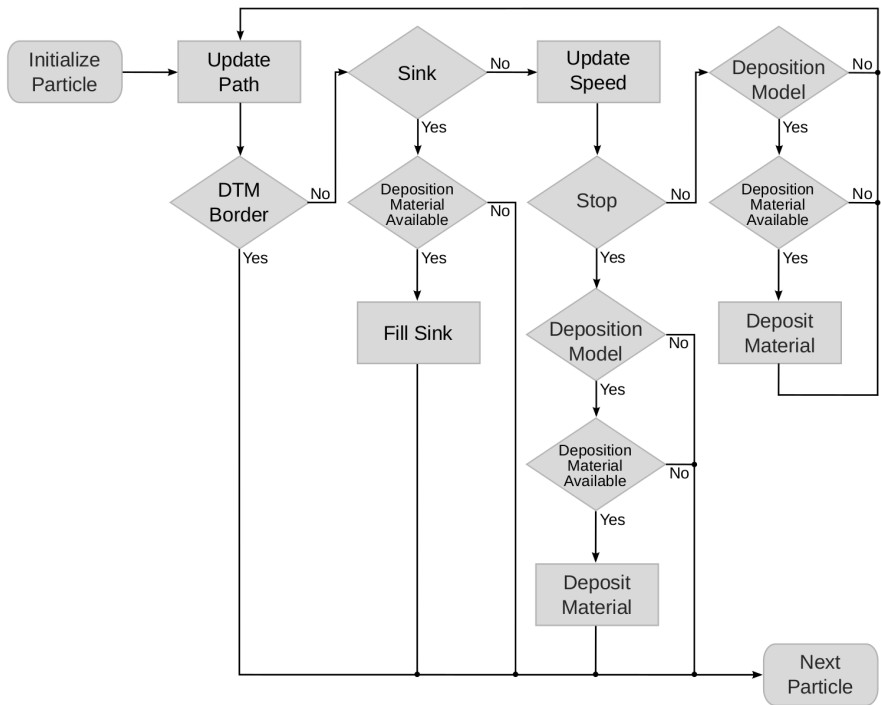

**Figure 3.** Flowchart of a fully featured GPP model configuration for scenario modeling.

10   The sequence in which release areas, respectively particles, are initialized is crucial in case material deposition is simulated.
The modification of the terrain between model iterations can influence process paths and run-out distances significantly. The
following processing orders are implemented:

(a) Release areas in sequence: the release areas are processed one by one; in each model iteration, all start cells of a release
area are processed in ascending order of their elevation. This configuration computes all model iterations for the start cells of
release area one, next for the start cells of release area two and so on.

(b) Release areas in sequence per iteration: the release areas are processed one by one in each model iteration; the start cells
are processed in ascending order of their elevation. This configuration computes a single model iteration with the start cells

of release area one, next with all start cells of release area two and so on; then the next model iteration is run over all release areas.

(c) Release areas in parallel per iteration: in each model iteration the start cells of all release areas are processed in ascending order of their elevation. With this configuration, all start cells are processed in each model iteration sorted by elevation, irrespective of their membership to a certain release area.

Depending on the overall configuration, the GPP model requires just a few or more parameters. These are either global parameters, used throughout the simulation, or (optionally) spatially distributed parameters provided as raster data sets. An example for the latter are spatially distributed friction values depending on factors like surface characteristics or water content.

## 3  Modeling approaches

Within the following sections, the modeling approaches currently implemented for each model component are described in detail. The user can choose which model should be used in each component and combine them to simulate various processes. Typical model configurations are presented in Sect. 4.

### 3.1  Process path modeling approaches

In order to determine the downslope process path of a particle from its initiation site, the GPP model implements two different approaches. One is a single flow direction algorithm, which selects that neighbor cell as next flow path cell to which the steepest downward slope is observed. The other, based on a random walk, is a multiple flow direction approach sensitive to the local slope conditions.

### 3.1.1  Maximum slope

This approach, as proposed by O'Callaghan and Mark (1984), is implemented mainly for convenience in order to provide a simple means to detect the process path along the gradient of gravity. A particle follows the steepest descent of the slope:

$$n = max\{(z - z_i)/d_i\} \tag{1}$$

where $n$ is the neighbor of steepest descent, $z$ is the elevation of the currently processed cell, $z_i$ is the elevation of neighbor cell $i$, and $d_i$ is the horizontal distance to neighbor cell $i$.

The model result is thus deterministic, with the exception of its behavior (as implemented in the GPP model) when two or more neighbor cells show the same steepest descent or when a flat area is reached. In the first case, one of the neighbors cells is chosen by random. On flat areas a set of potential neighbor cells is determined which is made up of all neighbors with the same elevation as the current cell which have not been traversed yet in the current model iteration. From this set, a process path cell is chosen by random. This introduces a probabilistic component. Further, the terrain could have been modified between two model iterations by sink filling or material deposition.

The *Maximum Slope* model approach has no special parameters besides those controlling the mode of operation of the GPP model main loop, like the number of model repetitions or the processing order. The pseudo-random number generator, used to choose a neighbor cell by random under the pre-described conditions, can be initialized either with the current time or a fixed seed value. The latter will always produce the same succession of values for a given seed value and will thus give the same results for consecutive tool runs.

### 3.1.2   Random walk

With this approach, the process path is modeled by a variant of the dfwalk model as proposed by Gamma (1996, 2000). It uses a stochastic way of path finding, which makes it possible to model the lateral spreading of a process by calculating several iterations from the same start position. Besides the parameters controlling the Monte Carlo simulation like the number of repetitions, the *Random Walk* approach has three parameters to calibrate the model in order to mimic the behavior of different
geomorphological processes: (i) a slope threshold controls below which terrain slope divergent flow is allowed; (ii) this is accompanied by an exponent for divergent flow: below the slope threshold, the parameter controls the degree of divergence; (iii) finally, a persistence factor can be used to preserve the direction of movement by weighting the current flow direction in order to account for inertia, which can be observed for debris flows or wet snow avalanches (Nohguchi, 1989; Takahashi et al., 1992). Rockfall may be modeled with (almost) no persistence and a higher degree of divergence.
For the currently processed grid cell, a set **N** of potential flow path cells is determined from all immediate neighbor cells in a 3 by 3 window, which have an equal or lower elevation than the central cell. This is done in several steps. First of all, for each neighbor cell $i$ a slope value $\gamma_i$, based on the slope threshold $\beta_{thres}$, is calculated (Gamma, 2000):

$$\gamma_i = \frac{\tan \beta_i}{\tan \beta_{thres}}, \qquad \beta_i \geq 0, \qquad i \in \{1, 2, ...8\} \tag{2}$$

where $\beta_i$ is the slope to neighbor cell $i$. The maximum value $\gamma_{max} = max\{\gamma_i\}$ is a measure on how close the slope to
the steepest neighbor is to the slope threshold. In case $\gamma_{max} > 1$ the set **N** of potential flow path cells is only made up of the steepest neighbor. Otherwise, the *mfdf* (multiple flow directions for debris flows; Gamma, 2000) criterion is used to decide which neighbor cells are additionally included in **N**:

$$\gamma_i \geq (\gamma_{max})^a \qquad (0 < \gamma_{max} \leq 1, \qquad a \geq 1) \tag{3}$$

where $a$ is the exponent to control the amount of divergent flow ($a \geq 1$). If $\gamma_i$ is greater than or equal to the *mfdf* criterion, then the neighbor $i$ is included in **N**. Thus, the set **N** is given by (Gamma, 2000):

$$\mathbf{N} = \{i \mid \gamma_i \geq (\gamma_{max})^a \quad \text{for } i \in \{1, 2, ...8\}\} \qquad \qquad \text{if } 0 < \gamma_{max} \leq 1 \text{ and} \tag{4a}$$
$$\mathbf{N} = \{i \mid \gamma_i = \gamma_{max} \quad \text{for } i \in \{1, 2, ...8\}\} \qquad \qquad \text{if } \gamma_{max} > 1 \tag{4b}$$

The slope threshold makes it possible to adjust the model to different topography: in steep sections of the process path, where the terrain slope is near the threshold, only steep neighbors are allowed in addition to the steepest descent. In flat sections, almost all lower neighbor cells are potential flow path cells and the tendency for divergent flow is increased. The degree of divergent flow below the slope threshold can be controlled by the exponent of divergent flow. This sensitivity to the terrain conditions is an important property which is missing in the modeling approaches developed for hydrological processes, which distribute the flow proportionally to the slope to all lower neighbors irrespective of the local topography (Gamma, 2000).

Finally, a cell is picked by random from the set $\mathbf{N}$. The probability for each cell $prob_i$ is given by

$$prob_i = \frac{f_i \cdot \tan \beta_i}{\sum_j f_j \cdot \tan \beta_j} \tag{5}$$

where $i$ describes the currently processed neighbor cell, $j$ depicts all neighbor cells in set $\mathbf{N}$, and $f$ is a weighting factor. In case the flow direction to neighbor $i$ equals the previous flow direction, $f$ equals the persistence factor $p$ (with $p \geq 1$), otherwise $f = 1$. A tendency to move towards the steepest descent is always given as the transition probabilities are weighted by slope. The persistence factor can be used to weight the current flow direction, which results in a higher probability that the neighbor in this direction gets selected. This property can be used to reduce abrupt changes in flow direction. Finally the transition probabilities are scaled to accumulated values between 0 and 1, and the pseudo-random generator is used to select one flow path cell from the set.

In the GPP model, the approach is extended to also handle flat areas. This is done as described for the *Maximum Slope* approach with the same restriction that a potential successor cell must not have been traversed yet in the current model iteration in order to prevent endless loops.

The result of several model iterations is a raster data set storing the transition frequencies, i.e. how many times a grid cell has been traversed. Figure 4 shows the effect of different parameter settings for the three calibration parameters slope threshold, exponent for divergent flow and persistence factor (the run-out length was calculated with the *Geometric Gradient* approach using an angle of 26.5°, see Sect. 3.2.1 (a)). The number of model iterations is set to 1000 in the examples (a) to (j). In Fig. 4 (a) to (e) the slope threshold (40°) and the persistence factor (1.0) are fixed, while the exponent for divergent flow is increased in several steps (1.0, 1.1, 1.2, 1.5, and 2.0). It is obvious that the extent of the process area increases significantly because of the higher degree of lateral spreading.

In Fig. 4 (f) to (j) the exponent for divergent flow (1.5) and the persistence factor (1.0) are fixed, while the slope threshold is increased gradually (15°, 20°, 30°, 40°, and 60°). It can be seen that the point at which lateral spreading is allowed is moving up the torrential fan, resulting in an increase of the total process area.

Figure 4 (k) to (o) shows the results of a stepwise increase of the persistence factor (1.0, 1.5, 2.0, 2.5, and 3.0) while the slope threshold (40°) and the exponent of divergent flow (2.0) are fixed. Here, only a single iteration was calculated from each start cell in order to visualize single trajectories. It is obvious that with higher persistence factors the number of changes in direction along a trajectory is decreasing.

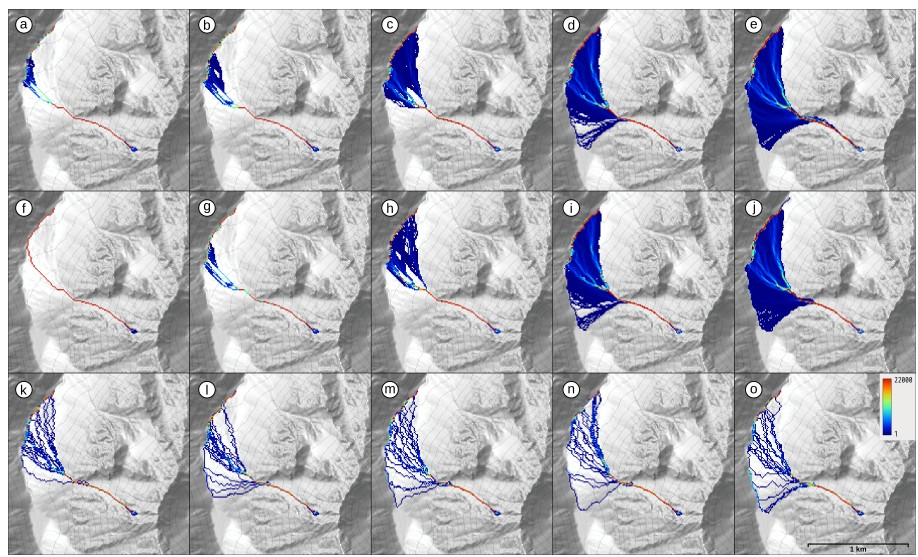

**Figure 4.** Effect of different random walk parameter settings; (a) to (e): different exponents for divergent flow (1.0, 1.1, 1.2, 1.5, and 2.0); (f) to (j): different slope thresholds (15°, 20°, 30°, 40°, and 60°); (k) to (o): different persistence factors (1.0, 1.5, 2.0, 2.5, and 3.0). For details see text.

## 3.2    Run-out modeling approaches

In order to determine the run-out length of a particle, several approaches are implemented in the GPP model. These range from rather simple but convenient approaches (regarding e.g., the comparison with field observations) based on the energy line principle to 1- and 2-parameter friction models. In the following, these approaches are described in detail.

### 3.2.1    Energy line approaches

The run-out length of a process is often described by the vertical and horizontal distances covered by a particle from its start to the stopping position:

$$\tan \alpha = dv/dh \tag{6}$$

where $\alpha$ is the angle to the horizontal and $dv$ and $dh$ are the vertical and horizontal offset, respectively. Both offsets can be defined differently, see below. This describes a straight energy line from the start to the stopping position (Heim, 1932). With a straight energy line, the velocity can be calculated by (Körner, 1980):

$$v_i = \sqrt{2 \cdot g \cdot h_v} \tag{7}$$

where $v_i$ is the velocity [ms$^{-1}$] on the currently processed grid cell, $g$ is the acceleration due to gravity [ms$^{-2}$] and $h_v$ is the height difference [m] between the energy line and the current grid cell $i$. Although the angle $\alpha$ is not constant, it can be observed that it has a characteristic value range for gravitational movements of a specific type. The calibration of the angle $\alpha$, which can be measured quite easily, is usually done by field observations and mapping. All approaches based on the energy line principle provide the possibility to output raster data sets storing the stopping positions and the maximum velocity encountered in each cell of the process path.

**(a) Geometric gradient:** The geometric gradient (Heim, 1932) defines the vertical offset as the vertical distance between the release area and the end of the deposit. The horizontal offset is defined as the horizontal distance between these two points. This modeling approach thus requires just the friction angle $\alpha$ as input. The GPP model supports both a global friction angle or a raster data set with friction angles for each start cell. Once the angle between the start cell of the particle and the current position of the particle drops below the friction angle $\alpha$, the end of the deposit is reached.

**(b) Fahrboeschung:** For the Fahrboeschung principle (Heim, 1932) the vertical offset is determined in the same way as for the geometric gradient. But the horizontal offset is not defined as the horizontal distance between start and end point but as the length of the horizontal projection of the actual process path. Again, the friction angle can be provided either as a global value or by a raster data set with friction angles for each start cell.

**(c) Shadow angle:** Both the geometric gradient and the Fahrboeschung principle do not take into account that with rockfalls most of the initial energy is dissipated once a rock impacts on the talus slope for the first time (Broilli, 1974; Dorren, 2003). Thus Hungr and Evans (1988) proposed the shadow angle, which defines the vertical offset as the vertical distance between the first impact location on the talus slope and the end of the deposit. The horizontal offset is defined as horizontal distance between the first impact location and the end of the deposit. From this it follows that the shadow angle is always smaller than the geometric gradient.

The shadow angle can again be provided either as a global value or by a raster data set with shadow angles for each start cell. In order to determine the location of the first impact of a particle on the talus slope, the GPP model implements two different approaches: (i) the user provides a raster data set with impact areas. Once a particle reaches a cell labeled as impact area, the location of this cell is used to measure the shadow angle; (ii) a threshold describing the slope angle above which free fall is assumed is provided. As soon as the angle between the start cell and the current position of the particle drops below the threshold, the location of this cell is used to measure the shadow angle.

### 3.2.2 1-parameter friction model

The 1-parameter friction model has been developed to simulate rockfall and is based upon concepts introduced by Scheidegger (1975), which have been extended by various authors (van Dijke and van Westen, 1990; Meißl, 1998; Dorren and Seijmonsbergen, 2003). The GPP model implements several of these approaches, more details can be found in Wichmann and Becht (2005) and Wichmann (2006). The 1-parameter friction model calculates the velocity on the currently processed grid cell according

to the velocity on the previous cell of the process path, the slope and a friction parameter. Once the velocity becomes zero, the end of the deposit is reached. Once a block is detached from the rock face, it is falling in free air:

$$v_i = \sqrt{2 \cdot g \cdot h_f} \tag{8}$$

where $v_i$ is the velocity [ms$^{-1}$] on the currently processed grid cell, $g$ is the acceleration due to gravity [ms$^{-2}$] and $h_f$ is the
height difference [m] between the start cell and the current grid cell $i$. The impact on the talus slope occurs, similar to the shadow angle model, if (a) a particle reaches a cell labeled as impact area or (b) the angle between the start cell and the current position of the particle drops below the free fall threshold. The decrease of velocity on the talus slope due to energy loss on the first impact can be calculated in two different ways:

(i) energy reduction (Scheidegger, 1975):

$$v_i = \sqrt{2 \cdot g \cdot h_f \cdot K} \tag{9}$$

where $K$ is the amount of unspent energy ($K \leq 1$, i.e. for an energy reduction of 75 % K is 0.25).

(ii) preserved component of velocity (Kirkby and Statham, 1975):

$$v_i = \sqrt{2 \cdot g \cdot h_f \cdot \sin \beta_i} \tag{10}$$

where $\beta_i$ denotes the local slope gradient [°]. Here, the component of the fall velocity parallel to the talus slope surface is
conserved.

Approach (i) requires the user to specify the amount of energy reduction as calibration parameter. With approach (ii) usually larger run-out distances are modeled. The strong dependence of approach (ii) on the slope of the impact cell complicates the model calibration (Wichmann, 2006). Approach (i) is used as the default in the GPP model. After the impact, two different modes of motion can be modeled (Scheidegger, 1975):
(i) sliding:

$$v_i = \sqrt{v_{(i-1)}^2 + 2 \cdot g \cdot (h - \mu_s \cdot D)} \tag{11}$$

where $v_{(i-1)}$ is the velocity [ms$^{-1}$] on the previous grid cell of the process path, $h$ is the height difference [m] between adjacent grid cells, $D$ is the horizontal difference [m] between adjacent grid cells and $\mu_s$ is the sliding friction coefficient [-].

(ii) rolling:

$$v_i = \sqrt{v_{(i-1)}^2 + 10/7 \cdot g \cdot (h - \mu_r \cdot D)} \tag{12}$$

where $\mu_r$ is the rolling friction coefficient [-].

Once the velocity on a grid cell becomes zero, the end of the deposit is reached. The model calibration usually requires only two parameters: the amount of energy loss on impact [%] and, depending on the chosen mode of motion, either the sliding or the rolling friction coefficient [-]. The friction coefficient can be provided as global value or spatially distributed by providing a raster data set with friction values. Impact on the talus slope can either be modeled by providing an input raster data set with

impact areas or by using a slope threshold (see Sect. 3.2.1 (c)). Besides the possibility to output a raster data set storing the stopping positions, a raster data set with the maximum velocity encountered in each cell of the process path can be output.

### 3.2.3   PCM model

The PCM model (Perla et al., 1980) is a 2-parameter friction model originally developed to calculate the run-out distance of avalanches. It is based on the model of Voellmy (1955). The model has also been applied to debris flows by various authors

(Rickenmann, 1990; Zimmermann et al., 1997; Gamma, 2000; Wichmann, 2006). It is a center of mass model and it is assumed that the motion is mainly governed by a sliding friction coefficient $\mu$ and a mass-to-drag ratio $M/D$. In steeper parts of the process path, the velocity is mainly influenced by $M/D$, whereas the velocity in the run-out area is dominated by $\mu$. The velocity on the currently processed grid cell depends on the velocity of the previous cell, the slope, the slope length and the two friction coefficients:

$$v_i = \sqrt{\alpha_i \cdot (M/D)_i \cdot \left(1 - e^{\beta_i}\right) + \left(v_{(i-1)}\right)^2 \cdot e^{\beta_i}} \qquad (13)$$

and

$$\alpha_i = g\left(\sin\theta_i - \mu_i \cos\theta_i\right)$$
$$\beta_i = \frac{-2L_i}{(M/D)_i} \qquad\qquad (14)$$

where $v_i$ is the velocity [ms$^{-1}$] on the currently processed grid cell, $g$ is the acceleration due to gravity [ms$^{-2}$], $\theta$ is the local slope [°], $L$ is the slope length between adjacent grid cells [m], $\mu$ is the sliding friction coefficient [-], and $M/D$ is the mass-to-drag ratio [m]. Perla et al. (1980) assume the following velocity correction for $v_{(i-1)}$ before $v_i$ is calculated in case of a concave transition in slope direction:

$$v^*_{(i-1)} = \begin{cases} v_{(i-1)} \cos\left(\theta_{(i-1)} - \theta_i\right) & \text{if } \theta_{(i-1)} \geq \theta_i \\ v_{(i-1)} & \text{if } \theta_{(i-1)} < \theta_i \end{cases} \qquad (15)$$

5      The correction is based on the conservation of linear momentum and has a higher magnitude in case of abrupt transitions. The accurate stopping position on a grid cell may be calculated by:

$$s = \frac{(M/D)_i}{2} \ln\left(1 - \frac{(v_{i-1})^2}{\alpha_i (M/D)_i}\right) \qquad (16)$$

where $s$ is the length [m] of the process path segment on the grid cell. In the GGP model, $s$ is not calculated and the process stops as soon as the square root in Eq. 13 becomes undefined. Thus the raster cell size determines the precision of the stopping position, which is a reasonable compromise for a grid based model.

Gamma (2000) proposed to incorporate the velocity correction (Eq. 15) directly into the velocity calculation (Eq. 13):

$$v_i = \sqrt{\alpha_i \cdot (M/D)_i \cdot (1 - e^{\beta_i}) + \left(v_{(i-1)}\right)^2 \cdot e^{\beta_i} \cdot \cos\left(\Delta\theta_i\right)} \tag{17}$$

and

$$\Delta\theta_i = \begin{cases} \theta_{(i-1)} - \theta_i & \text{if } \theta_{(i-1)} > \theta_i \\ 0 & \text{if } \theta_{(i-1)} \leq \theta_i \end{cases} \tag{18}$$

In the GPP model equation (17) is implemented. The model has to be calibrated by the friction parameters $\mu$ and $M/D$. In order to overcome the problem of mathematical redundancy – various combinations of the two parameters can result in the same run-out distance – the parameter $M/D$ is usually taken to be constant along the process path. It is only calibrated once in order to obtain realistic maximum velocity ranges for a given process. Both friction parameters can be provided either as a global value or spatially distributed by a raster data set. In the GPP model implementation it is also required to provide an initial velocity [ms$^{-1}$] in order to avoid that the process already stops on the first grid cell along the process path. As with the 1-parameter friction model, it is possible to output raster data sets storing the stopping positions and the maximum velocities.

### 3.3 Deposition modeling approaches

In the GPP model various deposition modeling approaches are implemented. In order to use these approaches, an input raster data set with material heights per start cell is required. This total material height is then averaged by the number of iterations to calculate the material height available for a particle in each iteration. Material that has not been spent in an iteration is made available for the remaining iterations. Deposited material immediately alters the terrain and the next iteration is computed on the modified DTM.

The most important deposition approach is the filling of sinks, which allows the GPP model to overcome small depressions or even larger obstacles like retention basins. Others simply deposit material once a particle stops or allow deposition along the process path based on slope and/or velocity thresholds. The latter can be used to model scenarios like the blocking of a channel by wood or debris.

### 3.3.1 Sink filling

The sink filling approach is immediately activated once a raster data set with material heights per start cell is provided as input. As soon as a sink is detected, the particle stops and material is deposited. The deposition is done preserving a downward slope if procurable, avoiding to create new sinks and making it possible to overcome the obstacle in subsequent model iterations.

The sink filling approach is based on Gamma (2000) with slight modifications: (i) the overflow cell and the depth of the sink are determined; (ii) if the depth of the sink can not be filled with the material available for the current model iteration, all material available is deposited and the computation stops; (iii) the sink is filled up to the height which is preserving a user specified minimum slope to the overflow cell; (iv) in order to avoid the creation of another sink, material is deposited on the process path above the sink; therefore it is tested if the material left is enough to fill up the process path above the sink while preserving the minimum slope; in case the available material is not enough to preserve this slope, the angle is continuously decreased until a minimal downward slope can be preserved. In case material is left, it is made available for the subsequent iterations. Gamma (2000) did not use a user specified minimum slope to preserve, but determined the average slope along the process path above the sink for the last 50 meters. In performance tests of the GPP model this turned out to be too dependent on the local slope conditions, often resulting in large angles and thus using too much material which is then missing to fill the sink upwards.

### 3.3.2   On stop

This approach simply deposits material on the grid cell of the modeled stopping position. The amount of material deposited on this cell is controlled by the *Initial Deposition on Stop* parameter, which describes the percentage of the available material which is deposited at the stopping position. The rest of the material is used to fill up the process path above the stopping position. The angle used to do this while preserving a downward slope is determined in a way that all material left in this iteration is used.

The approach makes it possible to adjust the deposition behavior to different geomorphological processes: simulating a rock fall event, the *Initial Deposition on Stop* parameter can be set to 100 %, resembling the deposition of single rocks. With debris flows or snow avalanches, it can be set lower in order to achieve a more lobe like deposition pattern. Nevertheless, the approach is not intended to realistically simulate the deposition pattern. But it can be used for scenario modeling, forcing the process path into different directions in subsequent model iterations.

### 3.3.3   Slope / velocity based

The *On Stop* deposition approach can be extended by slope and/or velocity based components, which can be used to force the deposition of material along the process path. Such components have been proposed by Gamma (2000) and are used in a modified way in the GPP model. Again, this approach is most useful for scenario modeling in order to simulate debris jamming or channel plugging. It is also useful if a high resolution DTM with great detail is used. The deposition starts once the slope or the velocity drops below a specific threshold. At a slope or velocity of zero, the *Maximum Deposition along Path* parameter controls the percentage of material (available in this model iteration) that is deposited. At the slope/velocity threshold the material deposition is zero, which results in a linear relation.

The slope and velocity based approaches can be used separately or in combination. In the latter case, a deposition height is calculated with both approaches and the lower deposition height is applied. This reduces artefacts resulting from the usage of a single threshold. For example, on flat areas, no material is deposited as long as the velocity is still high.

The slope/velocity based approaches have a further parameter, the *Minimum Path Length*, which describes the distance along the process path that must be exceeded before deposition sets in. This is required to simulate the behavior of a volume (and not single particles) and to prevent the deposition of material shortly after the process has initiated or even within the release area itself. It is also useful to have more control on the position along the process path where deposition should set in, especially in case of cascades with alternating steeper and gently dipping slope profile sections.

## 3.4 Model input and output

A brief summary on the GPP model parameters, input, and output data sets is given in the supplement: Table A1 shows the process path model parameters, grouped by model. The run-out parameters are shown in Table A2 and the deposition parameters in Table A3. Some of the parameters are global parameters, others can be provided as raster data sets in order to use spatially distributed parameter values. The input and output data sets are summarized in Table A4.

## 4 Model configurations and application examples

Use cases of the GPP model on a regional scale are natural hazard susceptibility mapping and the derivation of geomorphological process areas and sediment cascades. It is possible to simulate different scenarios based upon e.g., process magnitude, the existence of protection forest or protection measures. The inclusion of the deposition model component is usually only done on a more local scale. The modeling approaches available for each model component make it possible to simulate different gravitational processes depending on the overall model configuration. Within the following sections typical model configurations and parameter settings are described for rockfall, debris flow and avalanche modeling. Run-out calculations using one of the approaches based on the energy line principle have been used for all three process types, but as they are straight forward to use they are not discussed in detail. A separate section provides further information on scenario modeling. It must be noted that the parameter ranges provided for each process have to be considered as approximate values only and are thought to provide an initial guess. For example, Wichmann et al. (2008) have shown that for debris flow modeling the random walk and friction model parameters decrease with lower DTM resolutions.

## 4.1 Rockfall

A typical model configuration for rockfall modeling on a regional scale, e.g., to create susceptibility maps, combines the modeling approaches shown in Table 1. Usually the *Random Walk* approach is used to determine the process path, using rather permissive parameter settings regarding lateral spreading. The slope threshold is set rather high, usually in conformance with the threshold for free fall, in order to permit changes in direction already with the first impact on the talus slope. The exponent of divergence is comparatively high, too, in contrast to a rather small persistence factor which mimics the fact that rocks often change direction on impact.

The threshold of free fall used in the *1-parameter friction model* depends on the DTM resolution, but should conform with the slope threshold of the *Random Walk* model. The energy reduction on impact is usually about 75 % as investigated by

**Table 1.** Model configuration for rockfall modeling on a regional scale and approximate parameter ranges (compiled from Wichmann, 2006; Wichmann and Becht, 2006; Proske and Bauer, 2016).

| Model component | Model approach | Parameter | Value range |
|---|---|---|---|
| Process path | Random walk | slope threshold | 55–65° |
| | | exponent of divergence | 1.5–2.0 |
| | | persistence factor | 1.0–1.6 |
| Run-out | 1-parameter friction model | threshold free fall | 55–65° |
| | | energy reduction | 70–75 % |
| | | $\mu$ | 0.35–2.5, spatially distributed |
| | | mode of motion | sliding |

Broilli (1974). Although the dominating modes of motion of rockfalls are falling, bouncing, and rolling, often a sliding motion is simulated for the sake of simplicity (e.g., van Dijke and van Westen, 1990; Meißl, 1998; Dorren and Seijmonsbergen, 2003; Wichmann and Becht, 2005). When the model is applied on a regional scale, the friction coefficient $\mu$ should be provided spatially distributed as raster data set. Table 2 shows sliding friction coefficients for different materials and land cover. Spatially distributed friction coefficients are also very useful for scenario modeling, e.g., in order to determine the consequences of protection forest removal or reforestation.

The model configuration thus requires the following raster data sets as input: a DTM, a raster with release areas, and a raster with spatially distributed friction coefficients. Model outputs, describing the derived process area, are raster data sets storing the transition frequencies, the encountered maximum velocities and the stopping positions.

### 4.2 Debris flows

A typical model configuration for debris flow modeling on a regional scale is shown in Table 3. Again, the *Random Walk* approach is used for path finding. The slope threshold is usually set to angles slightly above the slope of the torrential fan. The exponent of divergence depends on the size of the simulated events. The larger the event, the higher the exponent. Its value also depends on the grain size and water content, with lower values for flowslides and higher values for coarse-grained debris flows. The persistence factor is higher compared to rockfall as persistence is given in the case of debris flows.

Run-out distances are calculated on basis of the PCM model. The $M/D$ drag ratio is usually calibrated once to match the highest observed velocities of a specific type of debris flow. The friction parameter $\mu$ is once again provided spatially distributed as a raster data set. Based on the observation that the sliding friction coefficient tends towards lower values with increasing catchment area, attributed to a changing rheology with higher discharges along the process path, Gamma (2000) derived the following estimating functions from debris flows in Switzerland:

minimum run-out:  $\mu = 0.25 \cdot a^{-0.21}$    likely run-out:  $\mu = 0.19 \cdot a^{-0.24}$    maximum run-out:  $\mu = 0.13 \cdot a^{-0.25}$

**Table 2.** Coefficients of friction for different materials and land cover (compiled from van Dijke and van Westen, 1990; Dorren and Seijmonsbergen, 2003; Wichmann, 2006).

| Material / land cover | Friction coefficients ($\mu$) |
| --- | --- |
| Tills | 0.35–0.5 |
| Residual soils | 0.4–0.5 |
| Fluvial materials | 0.4–0.5 |
| Bare rock | 0.4—0.9 |
| Scree materials: | |
| - marl | 0.35–0.45 |
| - flysch | 0.6–0.7 |
| - sandstone | 0.7–0.8 |
| - dolomite | 0.7–0.8 |
| - limestone | 0.8–0.9 |
| Rockfall materials | 0.9–1.0 |
| Meadow | 0.5–0.6 |
| Alpine shrubs | 0.6–0.9 |
| Bushes | 0.6–0.7 |
| Open forest | 1.0–2.0 |
| Dense forest | > 2.0 |

10    with $a$ = catchment area in km$^2$. Such data sets can be easily computed from a raster with stored catchment area (i.e. flow accumulation). Gamma (2000) and Wichmann and Becht (2005) additionally apply minimum (0.045) and maximum (0.3) thresholds in order to exclude extreme values. The model configuration thus requires a DTM, a raster with release areas, and a raster with spatially distributed friction coefficients as input. Model outputs, describing the derived process area, are again raster data sets storing transition frequencies, encountered maximum velocities and stopping positions.

## 15  4.3  Avalanches

The model configuration for avalanche modeling on a regional scale resembles that for debris flow modeling, but the parameter variability is higher because of the different properties of powder and wet snow avalanches (see Table 4). All *Random Walk* parameters usually require higher values in order to be able to reproduce the extent of the process area. The friction parameter $\mu$ is lower for larger events, and the lower the snow density is, with powder avalanches showing the lowest values. The $M/D$ ratio 20  is usually higher with larger (and powder) avalanches, resulting in higher maximum velocities. Both friction parameters can be provided spatially distributed. For example, Heckmann (2006) used spatially distributed $M/D$ values based on vegetation cover as substitute for surface roughness.

**Table 3.** Model configuration for debris flow modeling on a regional scale and approximate parameter ranges (compiled from Zimmermann et al., 1997; Gamma, 2000; Wichmann and Becht, 2005; Wichmann, 2006).

| Model component | Model approach | Parameter | Value range |
|---|---|---|---|
| Process path | Random walk | slope threshold | 20–40° |
| | | exponent of divergence | 1.3–3.0 |
| | | persistence factor | 1.5–2.0 |
| Run-out | PCM model | $\mu$ | 0.04–0.8, spatially distributed |
| | | M/D ratio | 20–150 |

**Table 4.** Model configuration for avalanche modeling on a regional scale and approximate parameter ranges (compiled from Perla et al., 1980; Salm et al., 1990; Hegg, 1996; Heckmann, 2006; Schmidtner, 2012).

| Model component | Model approach | Parameter | Value range |
|---|---|---|---|
| Process path | Random walk | slope threshold | 45–60° |
| | | exponent of divergence | 1.3–5.0 |
| | | persistence factor | 1.5–3.0 |
| Run-out | PCM model | $\mu$ | 0.1–0.5, spatially distributed |
| | | M/D ratio | 20–1000, spatially distributed |

## 4.4 Scenario modeling

Scenario modeling usually addresses topics like process magnitude, the impact of protection forest or protection measures. Different process magnitudes are usually modeled by using a different number of model iterations and/or friction coefficients. For example, different friction coefficients can be used to assess the relevance of protection forest by simulating events with and without forest cover and to compare how the run-out distances increase (e.g., Wichmann, 2006; Proske and Bauer, 2016). Different friction coefficients have also been used to simulate different block sizes in rockfall modeling (e.g., Haas et al., 2012b). The influence of protection measures can be analyzed by manipulating the DTM to include barriers or retention basins and to observe the impact on the extent of the processes area. Here, deposition modeling is usually involved for sink filling. Deposition of material and sink filling are also required with high resolution DTMs in order to fill up small depressions, to overcome obstacles, or to simulate the break out of incised channels.

In order to demonstrate the approach for sink filling, a 10 m DTM has been modified to include a sink along the process path. For the sake of simplicity, the process path is modeled using the *Maximum Slope* approach with 1000 iterations and no friction and deposition models. Figure 5 (a) shows that the process stops at the end of the sink in case no material is provided.

15  If 50 m$^3$ of material are provided, the process overcomes the sink and stops not until the next sink is reached. This sink can not be overcome because there is not enough material left.

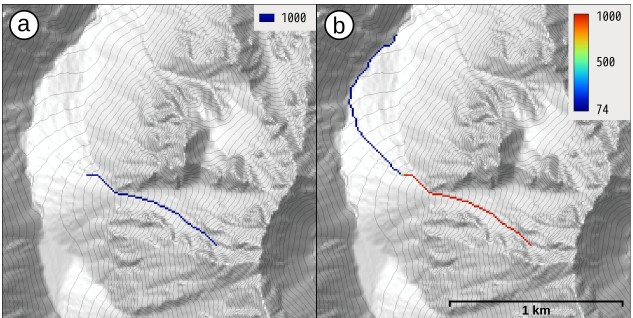

**Figure 5.** Sink filling; (a) the process stops in a sink; (b) the process overcomes the sink and stops in the next sink because no material is left.

Figure 6 illustrates the sink filling approach in detail. In case only a single iteration is calculated (Fig. 6 (a)), all material provided is available in that iteration. The sink can thus be filled at once, preserving the slope specified with the minimum slope parameter (here 2.5°). Figure 6 (b) shows the successive filling of the sink when ten model iterations are calculated and thus only 50/10 m$^3$ of material are available per iteration.

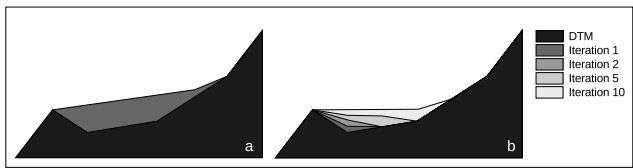

**Figure 6.** Longitudinal profile illustrating the sink filling approach; (a) single model iteration, (b) ten model iterations.

Figure 7 shows the result of modeling two different magnitudes of debris flow events from five release areas on a 10 m, hydrologically sound DTM. The process path is modeled with the *Random Walk* approach (slope threshold = 40°, exponent of
divergence = 2, persistence factor = 1.5, model iterations = 1000) and the run-out distance is calculated with the PCM model. Because debris flow velocities are usually lower than 12–15 ms$^{-1}$, $M/D$ is set to 40 m. The two events are modeled using a friction parameter $\mu$ of 0.25 for the medium event and a $\mu$ of 0.13 for the large event. In both cases the initial velocity is set to 1 ms$^{-1}$.

The maximum velocities reached along the steeper parts of the process path are almost the same (16 ms$^{-1}$ for the large event,
15 ms$^{-1}$ for the medium event), but the run-out distances significantly increase with the lower friction value $\mu$ used for the large event. The stopping positions are well distributed over the torrential fan because of the different process path lengths and slope profiles of the respective random walks. The number of stops per grid cell resembles the pattern of the transition frequencies.

Figures 8 (b) and (c) show the modeling results of the large event from four release areas on a hydrologically sound 2.5 m DTM (same random walk and friction model settings as in the 10 m case above). At this DTM resolution the debris flow

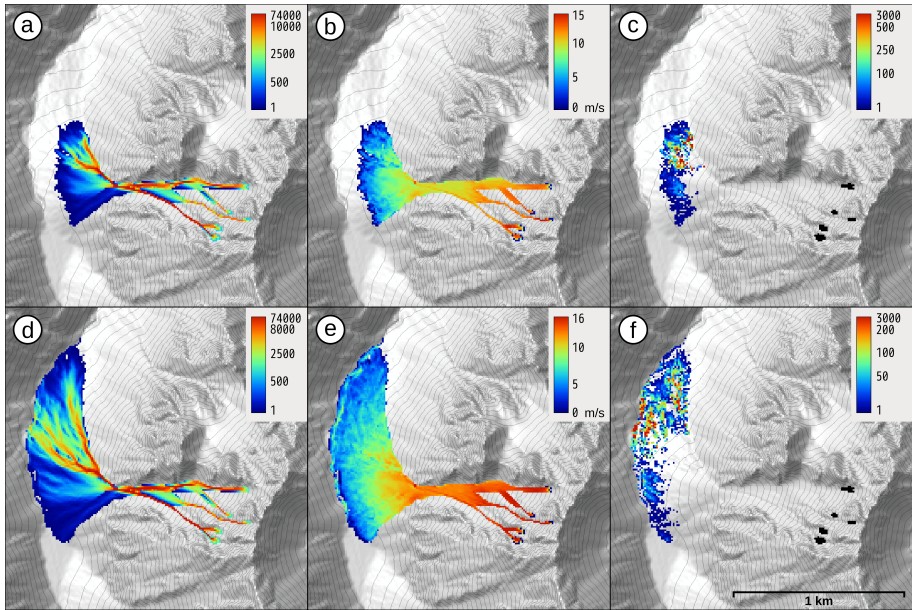

**Figure 7.** Medium (top) and large (bottom) debris flow events; (a) and (d) transition frequencies, (b) and (e) maximum velocities, and (c) and (f) stopping positions. For details see text.

channels are sharply incised and the process path is forced to follow the channels in case no material deposition along the process path is simulated. Figures 8 (d) to (f) show the result using 2750 m$^3$ of material in total (equally distributed over the release areas) and the deposition model approach *min(slope;velocity) & on stop* with the following parameter settings: initial deposition on stop = 20 %, slope threshold = 35°, velocity threshold = 12 ms$^{-1}$; maximum deposition along path = 20 % and minimum path length = 650 m. This parameter setting constrains the material deposition to the head of the torrential fan, successively filling up the incised channel and permitting the process to break out of the channel. In consequence, the process area covers the complete fan. Comparing the stopping positions (Fig. 8 (f)) with the material deposition heights (Fig. 8 (e)) it can be seen that although the deposition approach tries to deposit material while preserving a downward slope, new sinks are introduced in some cases because the available material per model iteration is not always enough to meet this requirement. Such sinks are then filled up in subsequent model iterations (see also Fig. 6 (b)). It can also be seen that all of the provided material is already used up before the end of the process paths is reached.

## 5   Discussion and Conclusion

The GPP model integrates several well known model approaches, which are established in practice into a single GIS-based simulation framework. The framework is highly modular, with components for process path, run-out length, sink filling and material deposition. The GPP model is a conceptual model, which provides the possibility to combine different modeling approaches and thus to model different kinds of gravitational processes. The currently implemented modeling approaches are

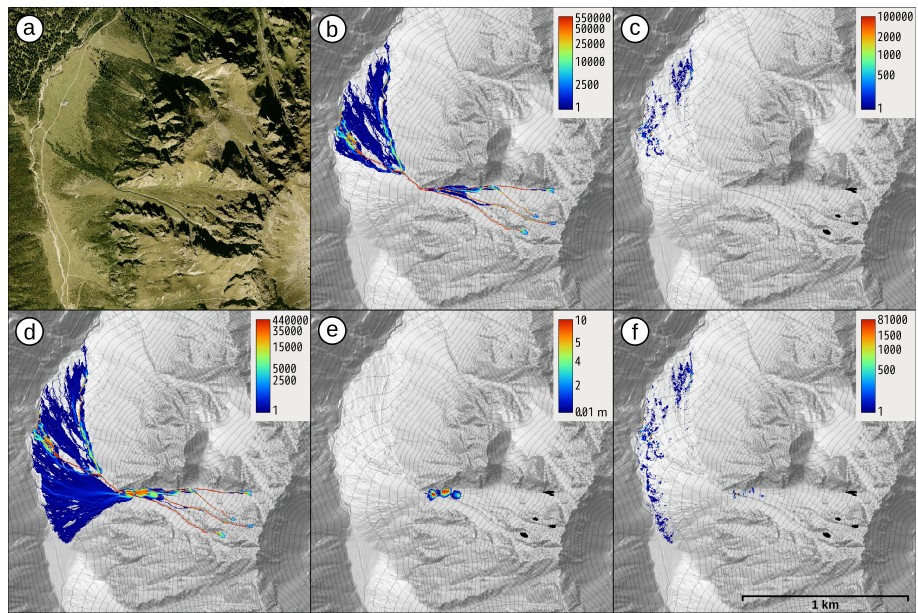

**Figure 8.** Deposition modeling scenario on a high resolution 2.5 m DTM; (a) orthophoto, (b) transition frequencies (no deposition), (c) stopping positions (no deposition), (d) transition frequencies (deposition), (e) material deposition heights, (f) stopping positions (deposition). For details see text.

not entirely physically based, but build on empirical and basic principles to mimic typical macroscopic characteristics of mass movements. Nowadays, several physically based numerical simulation models are available (e.g., Iverson, 1997; Pudasaini and Hutter, 2007), which make it possible to simulate processes at a very high level of precision. However, these types of models require many (geotechnical) parameters like rheological properties, cohesion and substrate characteristics. The detailed information required and the real-world heterogeneity limit their applicability to small areas, usually to single events (Clerici and Perego, 2000; Guthrie et al., 2008).

Although some modeling approaches included in the GPP model are based on rather simple concepts it is their complex interaction, which permits to delineate the extent of gravitational process areas. Reasonable results can be obtained with a minimum of input data and model parameters, recommending the framework especially for susceptibility mapping on regional scales. Recent additions like the model components for sink filling and deposition modeling make it also interesting for scenario modeling on various scales. Nevertheless, because of the limitations of the model it must be noted that this has to be done carefully at a local scale. For example, different block sizes of rockfall can only be simulated indirectly by using different friction parameters. Another limitation is the restriction of the process path routing to neighbor cells with equal or lower elevation, which makes the run-up of material on the opposite valley slope impossible. Like with every other simulation model it must be pointed out that it is a prerequisite to understand the functionality of the modeling approaches in detail before their application and the interpretation of the model results.

The GPP model provides only forward modeling capabilities. But as it is embedded in a GIS environment, model validation by observed historical events, e.g., by receiver operating characteristics (ROC curve), can be done outside the model. Also the derivation of initiation sites can be done within the GIS environment. Currently lacking are tools to automatically estimate model parameters based on observed process areas. This would be a great addition.

Frameworks for the simulation of gravitational mass movements on a regional scale have been released by various authors. For example, Horton et al. (2013) published the Flow-R (Flow path assessment of gravitational hazards at a Regional scale) model, which is a distributed empirical model for regional susceptibility assessments of debris flows. It includes several flow direction algorithms, but not all are relevant for gravitational mass movement modeling, and a random walk approach is missing. Flow-R also implements two friction models: the approach of Perla et al. (1980) and the simplified friction-limited model (SFLM), which is based on the Fahrboeschung principle (Heim, 1932). Flow-R is Matlab-based and available free of charge for Windows and Linux, but its source code is not open. Mergili et al. (2015) developed the r.randomwalk model which offers built-in functions for model validation and has the ability to consider uncertainties. It is a multi-functional conceptual tool for backward- and forward-analyses of mass movement propagation and implemented as add-on to GRASS GIS (but not officially included). It additionally requires the statistics software R (R Project for Statistical Computing). Currently the tool only works on UNIX systems with GRASS GIS 7.0 installed from source.

The GPP model is written in C++ and implemented in the "Geomorphology" tool library for the FOSS SAGA (Conrad et al., 2015). It is thus completely integrated into a GIS environment which facilitates the preparation of input data and the analysis of the results. This avoids cumbersome data editing and data format conversions. Furthermore, the integration of the model's source code into the official SAGA source code repository will assure source code maintenance and easy application since the GPP model will be included in every SAGA binary release. It is running on Windows, Linux and Mac OS X.

Besides its purely scientific application, the GPP model also qualifies as kind of sandbox game because of its characteristics. Dynamic processes are reproduced by stochastic components and Monte Carlo simulation. Basically only a DTM and a map of release areas is required to get started. This allows its straightforward application in education. Additional information like spatially distributed friction coefficients derived from land cover maps are easily added for scenario modeling. This allows for example to visualize the impact of protection forest decline on rockfall run-out length by simulating scenarios with and without forest cover through the application of different friction coefficients (see Table 2).

The GPP model is an attempt to bundle the development efforts put into several geomorphological process models within the last years into a single free and open source application. It is the author's opinion that making them available in a new and free implementation, even extended by new components, is important for geomorphological and natural hazards related research and education. The modular structure of the framework and in particular of the source code facilitates the addition of further model approaches. The author is looking forward to contributions like the extension of the framework through the addition of new modeling approaches or the implementation of accompanying SAGA tools, e.g., for automatic model parameter calibration based on observed events.

## 6 Code availability

The SAGA source code repository, including the GPP model, is hosted at https://sourceforge.net/projects/saga-gis/ using a git
repository. Read only access is possible without login. Alternatively, the source code and binaries can be downloaded directly
from the files section at https://sourceforge.net/projects/saga-gis/.

**Appendix A**

**Table A1.** The process path parameters of the GPP model.

| Model | Parameters | Description |
|---|---|---|
| Maximum slope | Iterations | Number of model iterations from each start cell [-] |
| | Processing order | Processing order of start cells; choice |
| | Seed value | Pseudo-random number generator initialization |
| Random walk | Iterations | Number of model iterations from each start cell [-] |
| | Processing order | Processing order of start cells; choice |
| | Seed value | Pseudo-random number generator initialization |
| | Slope threshold | Threshold below which lateral spreading is modeled [$°$] |
| | Exponent | Exponent controlling the amount of lateral spreading [-] |
| | Persistence factor | Factor used as weight for the current flow direction [-] |

**Table A2.** The run-out parameters of the GPP model.

| Model | Parameters | Description |
| --- | --- | --- |
| Geometric gradient | Friction angle | Angle between the release area and the end of the deposit (straight-line distance) [°]; either spatially distributed or global |
| Fahrboeschung principle | Friction angle | Angle between the release area and the end of the deposit (process path length) [°]; either spatially distributed or global |
| Shadow angle | Friction angle | Angle between first impact location on the talus slope and the end of the deposit (straight-line distance) [°]; either spatially distributed or global |
| | Threshold angle free fall | Minimum angle between start cell and current cell to model free fall [°]; alternatively a raster data set with slope impact areas can be provided |
| | Slope impact areas raster | Mapped slope impact areas as raster data set, optional |
| 1-parameter friction model | Threshold angle free fall | Minimum angle between start cell and current cell to model free fall [°]; alternatively a raster data set with slope impact areas can be provided |
| | Slope impact areas raster | Mapped slope impact areas as raster data set, optional |
| | Method impact | Approaches to calculate the velocity reduction on slope impact; choice |
| | Reduction | Amount of energy reduction on slope impact [%] |
| | Mu | Friction parameter $\mu$ [-]; alternatively a raster data set with friction values can be provided |
| | Mu raster | Spatially distributed friction values [-] as raster data set, optional |
| | Mode of motion | The mode of motion, either sliding or rolling |
| PCM model | Mu | Friction parameter $\mu$ [-]; alternatively a raster data set with friction values can be provided |
| | Mu raster | Spatially distributed friction values [-] as raster data set, optional |
| | Mass to drag ratio | Mass to drag ratio $M/D$ [m]; alternatively a raster data set with $M/D$ values can be provided |
| | Mass to drag ratio raster | Spatially distributed $M/D$ values [m] as raster data set, optional |
| | Initial velocity | The initial velocity of a particle [ms$^{-1}$] |

**Table A3.** The deposition parameters of the GPP model.

| Model | Parameters | Description |
|---|---|---|
| Sink Filling | Minimum slope | Minimum slope to preserve on sink filling [°] |
| On stop | Initial deposition on stop[1] | Percentage of available material initially deposited on stopping cell [%] |
| Slope & on stop | Slope threshold[2] | Slope angle below which the deposition of material sets in [°] |
| | Maximum deposition along process path[1] | Percentage of material which is deposited at most [%] |
| | Minimum path length[1] | Path length which has to be reached before material deposition is enabled [m] |
| Velocity & on stop | Parameters denoted by [1] | |
| | Velocity threshold | Velocity below which the deposition of material sets in [ms$^{-1}$] |
| min(slope;velocity) & on stop | Parameters denoted by [1,2] | |

[1]also used by the models below; [2]also used by the *min(slope;velocity) & on stop* model

**Table A4.** The input and output data sets of the GPP model.

| Data set | Description |
| --- | --- |
| Digital terrain model | In case no *Material* data set for sink filling is provided, this must be a hydrologically sound DTM [m]; input data set |
| Release Areas | Release areas labeled by unique integer IDs, all other cells NoData [-]; input data set |
| Material | Height of material available in each start cell [m]; used for sink filling and material deposition; optional input data set |
| Friction Angle | Spatially distributed friction angles [°]. Optionally used with the *Geometric Gradient*, *Fahrboeschung* or *Shadow Angle* friction model; optional input data set |
| Slope Impact Areas | Slope impact grid, impact areas labeled with valid values, all other NoData. Optionally used with the *Shadow Angle* or the *1-parameter friction model*; optional input data set |
| Friction Parameter Mu | Spatially distributed friction parameter $\mu$ [-], optionally used with the *1-parameter friction model* or the *PCM Model*; optional input data set |
| Mass to Drag Ratio | Spatially distributed $M/D$ ratio [m], optionally used with the *PCM Model*; optional input data set |
| Process Area | Delineated process area, stored as transition frequencies [count]; output data set |
| Deposition | Height of material deposited in each cell [m]; optional output data set in case a grid with material amounts is provided as input |
| Maximum Velocity | Maximum velocity observed in each cell [ms$^{-1}$]; optional output data set of the run-out models |
| Stopping Positions | Stopping positions, showing cells in which the run-out length has been reached [count]; optional output data set |

*Competing interests.* The author declares that he has no conflict of interest.

*Acknowledgements.* The author would like to thank the Federal State of Vorarlberg for providing the remote sensing data sets, especially P.
Drexel (Landesvermessungsamt Feldkirch).

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
