# Peer review of "The Gravitational Process Path (GPP) model (v1.0) – a GIS-based simulation framework for gravitational processes"

_Geoscientific Model Development, 2017_

## Referee Comment (RC1) · G. Meißl (Referee) · 16 Mar 2017

**Wichmann, V. (2017): The Gravitational Process Path (GPP) model (v1.0) – a GIS-based simulation framework for gravitational processes. GMD**

The paper presents a very valuable toolbox comprising established approaches which have partly been extended. The open source release of the GPP model supports simulating gravitational processes in university teaching as well as in engineering practice. The paper gives a scientifically sound presentation of the implemented components and can also serve as an extended manual for the GPP model. I strongly support the acceptance of the manuscript after some minor revisions.

**Detailed comments**

Page 1, line 5: I am not sure, if the expression "large-coverage studies" is common. I would suggest "studies covering large areas" or "regional scale studies".

Page 1, line 16/17: word repetition (working, work)

Page 2, line 20: "our" / Page 13, line 26: "for us"/ Page 21, line 3 and 5: "we": As the manuscript is presented by one author, it can be assumed from the citations of the precursor developments, who is meant with the plural expression "we". However, it may sound like a Pluralis Majestatis. I would recommend reformulating the sentences.

Page 2, line 24: Please give a short overview on the structure of the paper.

Page 2, line 26: "starting zones" might be better than "initiation sites".

Page 2, line 27: "represented" might be better than "organized".

Page 4, Figure 2: If the particle reaches the DTM border, there is no sink filling any more, I assume. Thus to my understanding, the arrow from DTM border should go directly to next particle and not lead to "Fill Sink".

Page 4, line 6: "combine them" instead of "combine these selections" as the components are combined, not the selections.

Page 5, introduction to chapter 3.1: The introduction to chapter 3.1 presents different approaches for process path modelling. It would help to avoid wrong expectations for the following subchapters if a last sentence could be added explaining which of the mentioned approaches are implemented within GPP and thus presented in detail. The introduction to chapter 3.1 is much longer than the introduction to chapter 3.2 (page 8). Maybe they can be homogenized regarding their length and structure.

Page 6, line 18: For me, the sentence would be more comprehensible if it would be: "This introduces a probabilistic component, especially if the terrain is modified by sink filling or material deposition between two model iterations."

Page 6, line 21-23: Please explain what the pseudo-random number generator is used for in the maximum slope approach.

Page 6, line 4: It is not absolutely necessary but would help if N were also defined N in words as the set of possible flow path cells.

Page 6, line 8: "different terrain" or "different topography" instead of "different relief"

Page 7, formula 4: To my opinion, the usage of j may lead to misunderstandings and is not necessary. It may be replaced by i.

Page 7, line 15-16: I don't understand the explanation within the brackets: which is also contained in the computation of the sum if i' belongs to N).

Page 7, line 14-18: N is the set of possible flow path cells, i' is the previous flow direction. Thus, strictly speaking, i' cannot be part of N, as a direction is not a cell. Of course, it can be understood that each element $n_i$ as part of N is attained from the previous cell by using a clearly defined direction. Nevertheless I would prefer a mathematically sound formulation.

Page 7, line 24-29: Bullet points for the description of the three parameters would enhance the readability of this passage.

Page 7, line 33/34: Please insert a reference to chapter 3.2.2 (where the Geometric gradient is explained).

Page 9: Geometric gradient, Fahrboeschung and Shadow angle are subchapters to 3.2.1 energy line approaches and should thus be numbered: 3.2.1.1, 3.2.1.2, 3.2.1.3

Page 9, line 26: Better "For the Fahrboeschung the vertical offset is determined like for the geometric gradient". (Passive, because the Fahrboeschung is a concept and cannot define anything, determine in order to avoid the repetition of the work define).

Page 10, line 28: Please use ≤ instead of <=.

Page 10, line 7: "Approach (i) requires the user to specify the amount of energy reduction in percent." After reading this sentence I asked myself how the user should do this and had to wait until page 11, line 19 for the explanation (that the value has to be calibrated). Maybe the first statement and the explanation can be brought together.

Page 11: From the example given later in the manuscript I understood that the decision if the sliding or rolling mode is calculated is not made by the model, but has to be taken by the model user. It would help if this could be noted here.

Page 12, line 24: Does "also" mean that formula 14 is implemented, too?

Page 13, line 22, line 24, line 32 /Page 18, line 18: "left" instead of "left over"

Page 13, line 3-4: word repetition (simulating)

Page 13, line 5: "achieve" instead of "archive"

Page 14, line 14: I am not sure if the expression "how many percent of the material" is correct.

Page 14, line 17: "artefacts" instead of "artifacts"

Page 15, Table 1 /Page 17, Table 3 und 4: please use "μ" instead of "mu".

Page 15, Table 1: The dominating processes within rockfalls are falling, bouncing and rolling. Sliding occurs only subordinatedly. However, it is not unusual to simulate rockfalls by using the sliding motion for the sake of simplicity, but should be discussed.

Page 18, chapter 4.4: Of course, GPP can be a valuable tool for investigating different scenarios and thus also estimating e.g. the impact of protection measures. However, due to the mostly conceptual run-out modelling approaches, it is very well suited at the regional scale and can only be used with extreme caution at the local scale. For example, different block sizes of rockfall can only be simulated indirectly by using different friction parameters. In order to avoid false expectations I would appreciate if the use restrictions at the local scale could be thematised at an appropriate place.

Page 19, line 2/3: 2.5 m - lease use a no-break space.

Page 20, Figure 8 e): Why are no deposition heights shown at the end of the paths?

Page 20, line 18: "extent" instead of "extend"

Page 20, line 18: "Reasonably" instead of "reasonable"

Page 21, line 2: "The GPP model should" might be better than "The GPP model is an attempt"

Table A.3: I would prefer to skip 1 and 2 and list all parameters completely. Of course this means a lot of repetitions, but makes it clearer.

---

## Referee Comment (RC2) · Anonymous Referee #2 · 3 May 2017

General comments

In his manuscript, V. Wichmann presents a GIS-based simulation framework for gravitational processes, i.e. a compilation of model components and their software implementation. It includes various well-known as well as recently developed approaches that conceptually or semi-physically represent displacement processes (but not initiation processes). This simulation framework can be of great interest and value to a broad range of academic, government and corporate users especially since its open-source implementation facilitates access and encourages customization.

While this paper is, in principle, worthy of publication in GMD, in my view the manuscript should still be substantially improved:

- The introductory section currently does not provide a general scientific background and motivation.

- The cited references are too narrowly focused on work by Wichmann and Heckmann, including non-refereed publications. The chosen methods / model components and general modelling approach needs to be situated in the broader context of (physically-based, conceptual and empirical) models of gravitational mass movements.

- The Discussion lacks depth. In particular, limitations are not discussed, and comparisons with similar models and software (including commercial products) are missing.

- The presentation of model structure and components in sections 2 and 3 should be partly re-arranged and re-written as it is often hard to follow (see detailed comments below).

I hope that the author will find these general comments as well as the following detailed comments useful in improving their manuscript.

Detailed comments

P1L12 This paper will attract the interest of a broader audience if it starts with a paragraph introducing the motivation for this work and the broader context, e.g. scientific and societal relevance and need for this kind of model and software implementation

P1L17 what is 2.5D in this context? perhaps too much detail for an introduction

P1L18 Rather than presenting what the author's GPP model is capable of, the author should first provide a brief overview of state-of-the-art modelling approaches for gravitational mass movements (including suitable references to the literature) and then indicate which of these approaches were chosen for / included in GPP and why

P1L20-21 A reference should be included to support this statement. 'simple' may be more appropriate than 'simplistic'

P1L23-P2L2 - Some of the Wichmann / Heckmann references are published in less accessible journals and conference proceedings which may overlap in content with some of the peer-reviewed publications by the same authors? A better selection from this set of papers plus additional relevant references to the work of other authors may be more appropriate here. As far as hazard susceptibility modelling is concerned, I noticed that statistical and machine-learning methods (e.g. logistic regression, generalized additive model, support vector machine), which are tremendously popular in this field, aren't mentioned.

P2L33 'the plugging of a channel' - check wording; perhaps 'clogged stream channels'?

P3L5-9 This information is too detailed for a 'general model structure' section. Focus on broad concepts and structures, and explain the general modelling approach. E.g. is the proposed model based on principles of physics or is this a more heuristic GIS-based approach, could it perhaps be referred to as a cellular automata model? The processing steps descript in P3L12-15 appear to suggest a more heuristic approach that certainly ensures mass preservation but is not capable of accommodating the physical (sliding / flowing) behaviour of solid to liquid mixtures of rock, soil, water, snow etc. that may be present in the various types of gravitational mass movements considered here. Such a simple approach may not necessarily be a bad thing, but the methodology should be contrasted appropriately with other possible modelling approaches, in particular physically based ones. In this context it appears to me that the approach presented here is similar to the cellular-automata model proposed by Guthrie et al. (2008) in Landslides in the narrower context of landslide modelling.

'particle' - In general, I have the feeling that this word may be misleading; at the very least, its meaning in the context of this model should be properly defined. To me the word 'particle' suggests that the model works with some elementary units of mass, e.g. 1 $m^3$ blocks, that are either passed on or deposited. Does the model really operate on such discrete elementary units, or does it determine amounts of material (e.g. 1.432 $m^3$, i.e. real values not multiples of discrete units)? - (In P4L8 particles are referred to as "start cells", which adds to the confusion, since a grid cell does not change its

location but particles are presumably passed along.)

P4L10-19 - What's the rationale behind these three strategies? What geomechanical process characteristics are they based on?

Figs. 1-3 - Are all three figures necessary, or does figure 3 contain all necessary information? Diamond shapes are used for decisions; while "sink" and "stop" may be (nearly) self-explanatory, I am having difficulties understanding why "Material" or "Deposition Model" would involve a yes/no decision. E.g. if the material "stops" (Stop: Yes), then it will be deposited in full - what additional decisions are necessary? Perhaps some re-wording might help, or slightly more detailed labelling of boxes.

P6, Equation (2): This equation is a bit hard to read at first because of the unusual use of a conditional statement within a set, i.e. two opening curly braces of same size. Also, $n\_i$ should just be $i$, the cell identifier; $n\_i$ has not been defined. Perhaps to separate equations (2a) and (2b) should be given for N for gamma_max <= 1 and gamma_max > 1. A less technical and more direct way of expressing this condition would be to say that all beta_i are smaller or equal beta_thres and that at least one adjacent grid cell is steeper than beta_thres, respectively. Perhaps a brief explanation should be given as to what process or principle this equation is based on. It seems that this explanation is provided in lines P724-30 - why not here, before entering into technical details?

P7L9 "only steep neighbors are allowed" - what is the physical meaning of this? Does this restriction represent a real dynamic process or is this an arbitrary modelling decision?

P7L11 "which is missing [in] the modeling approaches developed for hydrological processes" - insert "in"; provide reference

In Eq. (4), upper branch (i' \in N), it seems necessary to include the persistence factor p in the denominator, i.e. \sum_j{p tan\beta_j} rather than \sum_j{tan\beta}. In the numerator, placing the factor p before tan\beta_j would be preferable. With this, the

scaling mentioned in L19 seems avoidable as they would already add up to 1.

P7L18 "this property (Markov chain)" - the described property, which tries to represent inertia, doesn't seem to be related to the Markov property of stochastic processes.

P7L29-30 These statements concerning the dynamics properties of various types of mass movements should be supported with suitable references.

P7L33-38 run-out length calculated with geometric gradient approach - approach not introduced previously. In general, referring to section 3.2.1, how does this fit into the previously described framework that processes mass movements on a cell-by-cell basis while the run-out length approaches look at vectors from initiation points to potential run-out locations?

P8L15 This statement needs to be supported by a reference. The assumptions underlying the estimates presented in this paragraph should be outlined at least briefly.

P12 Eq. (16) The exponent $\beta_i$ shouldn't be in superscript when using the exponential function exp; just write $\exp\beta_i$ instead of $\exp^{\beta_i}$

Section 5 needs to discuss limitations of the presented model(s). E.g. this conceptual modelling approach is not entirely physically based; it builds upon basic principles such as mass preservation and tries to mimic typical macroscopic behaviours of various types of mass movements (e.g., divergence) without modelling the actual internal geomechanical dynamics (e.g. viscous flow etc. as applicable).

Run-up of material on the opposite valley slope doesn't seem to be possible in the GPP model since material is only transferred to lower-elevation neighbouring cells as mentioned in P6L27.

P21L14 "impact of ... immediately obvious" - This is not a result of this study. The authors should avoid claims that are neither based on their findings nor on the cited literature. It may be appropriate to state that the proposed model can be used to assess the potential effectiveness of such forests (provided that the relevant processes are

adequately represented by this model and its parameters).

A very brief section outlining implementation details should be included. E.g. which programming language; parallelized implementation? how parallelized, e.g. different Monte Carlo repetitions executed in parallel, ...?

Is it correct that this model implementation only provides forward modelling capabilities, i.e. modelling possible outcomes based on prescribed model parameters? Or is it also capable of estimating model parameters such as the persistence parameter based on observed runout distributions? Are there any capabilities for validating the model based on observed runout distributions, or does the user have to do this outside the GPP module? I am thinking AUROC estimation based on observed historical events as commonly done in the statistical landslide susceptibility modelling literature.

The presented model and its implementation should be contrasted against other models and software, including commercial products, at least at a general level.

If I understand correctly the present model does not implement scouring (erosion) along the path as implemented by other similar models such as Guthrie et al. (2008) in Landslides.

P21 Section 6 - This shouldn't be a separate section (unless required by journal policy)

Technical comments

P1L7 "practicability" -> applicability or feasibility

P1L8-9 "first ... re-written" seems contradictory; it can't be first if it has been re-written, extended and improved

P1 'large-coverage...' -> 'regional-scale'

P2L26 'disposition modelling ' - susceptibility or slope stability modelling?

P7L21 "like" -> "as"

P7L24 "iterations" -> "repetition"

P7L29 "fixation" - re-word

P9L9 (i.e. first line) and elsewhere: "encoded" - re-word

P20L15 "proven" - re-word

P21L9 "pure" -> "purely"

---

## Author Comment (AC1) · 30 May 2017

**Author's response**

"The Gravitational Process Path (GPP) model (v1.0) – a GIS-based simulation framework for gravitational processes" by Volker Wichmann

Dear Getraud Meißl,

thank you very much for your valuable comments on the discussion paper. They were very helpful in order to improve the manuscript. I've used your comments and suggestions to rework the paper in many places, please have a look at the detailed responses below.

This response is structured as follows: the first section (as required by journal policy) includes your comments. In section two I have added my responses to the comments. Finally a document which highlights the actual changes made to the document is attached.

Thanks again and best regards,

Volker Wichmann

**1) Comments from referee 1**

Wichmann, V. (2017): The Gravitational Process Path (GPP) model (v1.0) – a GIS-based simulation framework for gravitational processes. GMD

The paper presents a very valuable toolbox comprising established approaches which have partly been extended. The open source release of the GPP model supports simulating gravitational processes in university teaching as well as in engineering practice. The paper gives a scientifically sound presentation of the implemented components and can also serve as an extended manual for the GPP model. I strongly support the acceptance of the manuscript after some minor revisions.

Detailed comments

Page 1, line 5: I am not sure, if the expression "large-coverage studies" is common. I would suggest "studies covering large areas" or "regional scale studies".

Page 1, line 16/17: word repetition (working, work)

Page 2, line 20: "our" / Page 13, line 26: "for us"/ Page 21, line 3 and 5: "we": As the manuscript is presented by one author, it can be assumed from the citations of the precursor developments, who is meant with the plural expression "we". However, it may sound like a Pluralis Majestatis. I would recommend reformulating the sentences.

Page 2, line 24: Please give a short overview on the structure of the paper.

Page 2, line 26: "starting zones" might be better than "initiation sites".

Page 2, line 27: "represented" might be better than "organized".

Page 4, Figure 2: If the particle reaches the DTM border, there is no sink filling any more, I assume. Thus to my understanding, the arrow from DTM border should go directly to next particle and not lead to "Fill Sink".

Page 4, line 6: "combine them" instead of "combine these selections" as the components are combined, not the selections.

Page 5, introduction to chapter 3.1: The introduction to chapter 3.1 presents different approaches for process path modelling. It would help to avoid wrong expectations for the following subchapters if a last sentence could be added explaining which of the mentioned approaches are implemented within GPP and thus presented in detail. The introduction to chapter 3.1 is much longer than the introduction to chapter 3.2 (page 8). Maybe they can be homogenized regarding their length and structure.

Page 6, line 18: For me, the sentence would be more comprehensible if it would be: "This introduces a probabilistic component, especially if the terrain is modified by sink filling or material deposition between two model iterations."

Page 6, line 21-23: Please explain what the pseudo-random number generator is used for in the maximum slope approach.

Page 6, line 4: It is not absolutely necessary but would help if N were also defined N in words as the set of possible flow path cells.

Page 6, line 8: "different terrain" or "different topography" instead of "different relief"

Page 7, formula 4: To my opinion, the usage of j may lead to misunderstandings and is not necessary. It may be replaced by i.

Page 7, line 15-16: I don't understand the explanation within the brackets: which is also contained in the computation of the sum if i' belongs to N).

Page 7, line 14-18: N is the set of possible flow path cells, i' is the previous flow direction. Thus, strictly speaking, i' cannot be part of N, as a direction is not a cell. Of course, it can be understood that each element n i as part of N is attained from the previous cell by using a clearly defined direction. Nevertheless I would prefer a mathematically sound formulation.

Page 7, line 24-29: Bullet points for the description of the three parameters would enhance the readability of this passage.

Page 7, line 33/34: Please insert a reference to chapter 3.2.2 (where the Geometric gradient is explained).

Page 9: Geometric gradient, Fahrboeschung and Shadow angle are subchapters to 3.2.1 energy line approaches and should thus be numbered: 3.2.1.1, 3.2.1.2, 3.2.1.3

Page 9, line 26: Better "For the Fahrboeschung the vertical offset is determined like for the geometric gradient". (Passive, because the Fahrboeschung is a concept and cannot define anything, determine in order to avoid the repetition of the work define).

Page 10, line 28: Please use $\leq$ instead of <=.

Page 10, line 7: "Approach (i) requires the user to specify the amount of energy reduction in percent." After reading this sentence I asked myself how the user should do this and had to wait until page 11, line 19 for the explanation (that the value has to be calibrated). Maybe the first statement and the explanation can be brought together.

Page 11: From the example given later in the manuscript I understood that the decision if the sliding or rolling mode is calculated is not made by the model, but has to be taken by the model user. It would help if this could be noted here.

Page 12, line 24: Does "also" mean that formula 14 is implemented, too?

Page 13, line 22, line 24, line 32 /Page 18, line 18: "left" instead of "left over"

Page 13, line 3-4: word repetition (simulating)

Page 13, line 5: "achieve" instead of "archive"

Page 14, line 14: I am not sure if the expression "how many percent of the material" is correct.

Page 14, line 17: "artefacts" instead of "artifacts"

Page 15, Table 1 /Page 17, Table 3 und 4: please use "μ" instead of "mu".

Page 15, Table 1: The dominating processes within rockfalls are falling, bouncing and rolling. Sliding occurs only subordinatedly. However, it is not unusual to simulate rockfalls by using the sliding motion for the sake of simplicity, but should be discussed.

Page 18, chapter 4.4: Of course, GPP can be a valuable tool for investigating different scenarios and thus also estimating e.g. the impact of protection measures. However, due to the mostly conceptual run-out modelling approaches, it is very well suited at the regional scale and can only be used with extreme caution at the local scale. For example, different block sizes of rockfall can only be simulated indirectly by using different friction parameters. In order to avoid false expectations I would appreciate if the use restrictions at the local scale could be thematised at an appropriate place.

Page 19, line 2/3: 2.5 m - lease use a no-break space.

Page 20, Figure 8 e): Why are no deposition heights shown at the end of the paths?

Page 20, line 18: "extent" instead of "extend"

Page 20, line 18: "Reasonably" instead of "reasonable"

Page 21, line 2: "The GPP model should" might be better than "The GPP model is an attempt"

Table A.3: I would prefer to skip 1 and 2 and list all parameters completely. Of course this means a lot of repetitions, but makes it clearer.

**2. Author's response to referee 1:**

Wichmann, V. (2017): The Gravitational Process Path (GPP) model (v1.0) – a GIS-based simulation framework for gravitational processes. GMD

"The paper presents a very valuable toolbox comprising established approaches which have partly been extended. The open source release of the GPP model supports simulating gravitational processes in university teaching as well as in engineering practice. The paper gives a scientifically sound presentation of the implemented components and can also serve as an extended manual for the GPP model. I strongly support the acceptance of the manuscript after some minor revisions."

Detailed comments

"Page 1, line 5: I am not sure, if the expression "large-coverage studies" is common. I would suggest "studies covering large areas" or "regional scale studies"."

**Response:** edited

"Page 1, line 16/17: word repetition (working, work)"

**Response:** I've removed the sentence completely because this was also addressed by Reviewer 2

"Page 2, line 20: "our" / Page 13, line 26: "for us"/ Page 21, line 3 and 5: "we": As the manuscript is presented by one author, it can be assumed from the citations of the precursor developments, who is meant with the plural expression "we". However, it may sound like a Pluralis Majestatis. I would recommend reformulating the sentences."

**Response:** rephrased

"Page 2, line 24: Please give a short overview on the structure of the paper."

**Response:** added

"Page 2, line 26: "starting zones" might be better than "initiation sites"."

**Response:** added

"Page 2, line 27: "represented" might be better than "organized"."

**Response:** I've kept "organized" since the release areas are an implementation concept in the GPP model which groups individual start cells into release areas

"Page 4, Figure 2: If the particle reaches the DTM border, there is no sink filling any more, I assume.

Thus to my understanding, the arrow from DTM border should go directly to next particle and not lead to "Fill Sink"."

**Response:** of course you are right, corrected

"Page 4, line 6: "combine them" instead of "combine these selections" as the components are combined, not the selections."

**Response:** corrected

"Page 5, introduction to chapter 3.1: The introduction to chapter 3.1 presents different approaches for process path modelling. It would help to avoid wrong expectations for the following subchapters if a last sentence could be added explaining which of the mentioned approaches are implemented within GPP and thus presented in detail. The introduction to chapter 3.1 is much longer than the introduction to chapter 3.2 (page 8). Maybe they can be homogenized regarding their length and structure."

**Response:** I've restructured the introduction to chapter 3.1 to match that of chapter 3.2 and moved parts of the introduction of chapter 3.1 to the discussion section

"Page 6, line 18: For me, the sentence would be more comprehensible if it would be: "This introduces a probabilistic component, especially if the terrain is modified by sink filling or material deposition between two model iterations.""

**Response:** rephrased

"Page 6, line 21-23: Please explain what the pseudo-random number generator is used for in the maximum slope approach."

**Response:** explanation added

"Page 6, line 4: It is not absolutely necessary but would help if N were also defined N in words as the set of possible flow path cells."

**Response:** N is already defined above (page 6, line 26) as "a set N of potential flow path cells"; this part of the manuscript has been reworked, including the equation

"Page 6, line 8: "different terrain" or "different topography" instead of "different relief""

**Response:** changed to "different topography"

"Page 7, formula 4: To my opinion, the usage of j may lead to misunderstandings and is not necessary. It may be replaced by i."

**Response:** replaced by i; the formula has been completely reworked, see next responses

"Page 7, line 15-16: I don't understand the explanation within the brackets: which is also contained in the computation of the sum if i' belongs to N)."

**Response:** the formula has been completely reworked by adding a weighting factor which is 1.0 or the persistence factor

"Page 7, line 14-18: N is the set of possible flow path cells, i' is the previous flow direction. Thus, strictly speaking, i' cannot be part of N, as a direction is not a cell. Of course, it can be understood that each element n i as part of N is attained from the previous cell by using a clearly defined direction. Nevertheless I would prefer a mathematically sound formulation."

**Response:** the formula has been completely reworked by adding a weighting factor and i' has been removed

"Page 7, line 24-29: Bullet points for the description of the three parameters would enhance the readability of this passage."

**Response:** readability improved by adding (i), (ii), and (iii)

"Page 7, line 33/34: Please insert a reference to chapter 3.2.2 (where the Geometric gradient is explained)."

**Response:** rephrased and reference added

"Page 9: Geometric gradient, Fahrboeschung and Shadow angle are subchapters to 3.2.1 energy line approaches and should thus be numbered: 3.2.1.1, 3.2.1.2, 3.2.1.3"

**Response:** the manuscript preparation guide lines allow only three levels of sectioning, so this was considered to be the best compromise

"Page 9, line 26: Better "For the Fahrboeschung the vertical offset is determined like for the geometric gradient". (Passive, because the Fahrboeschung is a concept and cannot define anything, determine in order to avoid the repetition of the work define)."

**Response:** rephrased

"Page 10, line 28: Please use $\leq$ instead of <=."

**Response:** changed

"Page 10, line 7: "Approach (i) requires the user to specify the amount of energy reduction in percent." After reading this sentence I asked myself how the user should do this and had to wait until page 11, line 19 for the explanation (that the value has to be calibrated). Maybe the first statement and the explanation can be brought together."

**Response:** rephrased, explanation added

"Page 11: From the example given later in the manuscript I understood that the decision if the sliding or rolling mode is calculated is not made by the model, but has to be taken by the model user. It would help if this could be noted here."

**Response:** rephrased to make clear that also the mode of motion has to be chosen by the user

"Page 12, line 24: Does "also" mean that formula 14 is implemented, too?"

**Response:** rephrased in order to make clear that only equation 16 is implemented, similar to Gamma (2000)

"Page 13, line 22, line 24, line 32 /Page 18, line 18: "left" instead of "left over""

**Response:** changed throughout the document

"Page 13, line 3-4: word repetition (simulating)"

**Response:** rephrased

"Page 13, line 5: "achieve" instead of "archive""

**Response:** corrected

"Page 14, line 14: I am not sure if the expression "how many percent of the material" is correct."

**Response:** rephrased

"Page 14, line 17: "artefacts" instead of "artifacts""

**Response:** changed

"Page 15, Table 1 /Page 17, Table 3 und 4: please use "μ" instead of "mu"."

**Response:** changed

"Page 15, Table 1: The dominating processes within rockfalls are falling, bouncing and rolling. Sliding occurs only subordinatedly. However, it is not unusual to simulate rockfalls by using the sliding motion for the sake of simplicity, but should be discussed."

**Response:** explanation and references added

"Page 18, chapter 4.4: Of course, GPP can be a valuable tool for investigating different scenarios and thus also estimating e.g. the impact of protection measures. However, due to the mostly conceptual run-out modelling approaches, it is very well suited at the regional scale and can only be used with extreme caution at the local scale. For example, different block sizes of rockfall can only be simulated indirectly by using different friction parameters. In order to avoid false expectations I would appreciate if the use restrictions at the local scale could be thematised at an appropriate place."

**Response:** addressed and added to the discussion

"Page 19, line 2/3: 2.5 m - lease use a no-break space."

**Response:** done, although that might be needed elsewhere in the final layout too

"Page 20, Figure 8 e): Why are no deposition heights shown at the end of the paths?"

**Response:** This is because all material available is already deposited before reaching the end of the process paths, i.e. there is not material left; explanation added to the text

"Page 20, line 18: "extent" instead of "extend""

**Response:** corrected

"Page 20, line 18: "Reasonably" instead of "reasonable""

**Response:** rephrased as "reasonable results" was meant

"Page 21, line 2: "The GPP model should" might be better than "The GPP model is an attempt""

**Response:** left unchanged

"Table A.3: I would prefer to skip 1 and 2 and list all parameters completely. Of course this means a lot of repetitions, but makes it clearer."

**Response:** left unchanged; besides a lot of repetitions this was mainly done because of layout considerations as the table gets quite large; but this could be changed easily (actually that was my first version, still commented in the latex sources)

**3. Manuscript with marked changes**

[revised manuscript text omitted]

---

## Author Comment (AC2) · 30 May 2017

**General comments**

In his manuscript, V. Wichmann presents a GIS-based simulation framework for gravitational processes, i.e. a compilation of model components and their software implementation. It includes various well-known as well as recently developed approaches that conceptually or semi-physically represent displacement processes (but not initiation processes). This simulation framework can be of great interest and value to a broad range of academic, government and corporate users especially since its open-source implementation facilitates access and encourages customization.

While this paper is, in principle, worthy of publication in GMD, in my view the manuscript should still be substantially improved:

- The introductory section currently does not provide a general scientific background and motivation.

- The cited references are too narrowly focused on work by Wichmann and Heckmann, including nonrefereed publications. The chosen methods / model components and general modelling approach needs to be situated in the broader context of (physically-based, conceptual and empirical) models of gravitational mass movements.

- The Discussion lacks depth. In particular, limitations are not discussed, and comparisons with similar models and software (including commercial products) are missing.

- The presentation of model structure and components in sections 2 and 3 should be partly re-arranged and re-written as it is often hard to follow (see detailed comments below).

I hope that the author will find these general comments as well as the following detailed comments useful in improving their manuscript.

**Detailed comments**

P1L12 This paper will attract the interest of a broader audience if it starts with a paragraph introducing the motivation for this work and the broader context, e.g. scientific and societal relevance and need for this kind of model and software implementation

P1L17 what is 2.5D in this context? perhaps too much detail for an introduction

P1L18 Rather than presenting what the author's GPP model is capable of, the author should first provide a brief overview of state-of-the-art modelling approaches for gravitational mass movements (including suitable references to the literature) and then indicate which of these approaches were chosen for / included in GPP and why

P1L20-21 A reference should be included to support this statement. 'simple' may be more appropriate than 'simplistic'

P1L23-P2L2 - Some of the Wichmann / Heckmann references are published in less accessible journals and conference proceedings which may overlap in content with some of the peer-reviewed publications by the same authors? A better selection from this set of papers plus additional relevant references to the work of other authors may be more appropriate here.

P1L23-P2L2 As far as hazard susceptibility modelling is concerned, I noticed that statistical and machine-learning methods (e.g. logistic regression, generalized additive model, support vector machine), which are tremendously popular in this field, aren't mentioned.

P2L33 'the plugging of a channel' - check wording; perhaps 'clogged stream channels'?

P3L5-9 This information is too detailed for a 'general model structure' section. Focus on broad concepts and structures, and explain the general modelling approach. E.g. is the proposed model based on principles of physics or is this a more heuristic GIS-based approach, could it perhaps be referred to as a cellular automata model? The processing steps descript in P3L12-15 appear to suggest a more heuristic approach that certainly ensures mass preservation but is not capable of accommodating the physical (sliding / flowing) behaviour of solid to liquid mixtures of rock, soil, water, snow etc. that may be present in the various types of gravitational mass movements considered here. Such a simple approach may not necessarily be a bad thing, but the methodology should be contrasted appropriately with other possible modelling approaches, in particular physically based ones. In this context it appears to me that the approach presented here is similar to the cellular-automata model proposed by Guthrie et al. (2008) in Landslides in the narrower context of landslide modelling.

P3L5-9 'particle' - In general, I have the feeling that this word may be misleading; at the very least, its meaning in the context of this model should be properly defined. To me the word 'particle' suggests that the model works with some elementary units of mass, e.g. 1 m 3 blocks, that are either passed on or deposited. Does the model really operate on such discrete elementary units, or does it determine amounts of material (e.g. 1.432 m 3, i.e. real values not multiples of discrete units)? - (In P4L8 particles are referred to as "start cells", which adds to the confusion, since a grid cell does not change its location but particles are presumably passed along.)

P4L10-19 - What's the rationale behind these three strategies? What geomechanical process characteristics are they based on?

Figs. 1-3 - Are all three figures necessary, or does figure 3 contain all necessary information? Diamond shapes are used for decisions; while "sink" and "stop" may be (nearly) self-explanatory, I am having difficulties understanding why "Material" or "Deposition Model" would involve a yes/no decision. E.g. if the material "stops" (Stop: Yes), then it will be deposited in full - what additional decisions are necessary? Perhaps some re-wording might help, or slightly more detailed labelling of boxes.

P6, Equation (2): This equation is a bit hard to read at first because of the unusual use of a conditional statement within a set, i.e. two opening curly braces of same size. Also, n\_i should just be i, the cell identifier; n\_i has not been defined. Perhaps to separate equations (2a) and (2b) should be given for N for gamma\_max <= 1 and gamma\_max > 1. A less technical and more direct way of expressing this condition would be to say that all beta\_i are smaller or equal beta\_thres and that at least one adjacent grid cell is steeper than beta\_thres, respectively. Perhaps a brief explanation should be given as to what process or principle this equation is based on. It seems that this explanation is provided in lines P724-30 - why not here, before entering into technical details?

P7L9 "only steep neighbors are allowed" - what is the physical meaning of this? Does this restriction represent a real dynamic process or is this an arbitrary modelling decision?

P7L11 "which is missing [in] the modeling approaches developed for hydrological processes" - insert "in"; provide reference

In Eq. (4), upper branch (i' \in N), it seems necessary to include the persistence factor p in the denominator, i.e. \sum\_j{p tan\beta\_j} rather than \sum\_j{tan\beta}. In the numerator, placing the factor p before tan\beta\_j would be preferable. With this, the scaling mentioned in L19 seems avoidable as they would already add up to 1.

P7L18 "this property (Markov chain)" - the described property, which tries to represent inertia, doesn't seem to be related to the Markov property of stochastic processes.

P7L29-30 These statements concerning the dynamics properties of various types of mass movements should be supported with suitable references.

P7L33-38 run-out length calculated with geometric gradient approach - approach not introduced previously. In general, referring to section 3.2.1, how does this fit into the previously described framework that processes mass movements on a cell-by-cell basis while the run-out length approaches look at vectors from initiation points to potential run-out locations?

P8L15 This statement needs to be supported by a reference. The assumptions underlying the estimates presented in this paragraph should be outlined at least briefly.

P12 Eq. (16) The exponent \beta\_i shouldn't be in superscript when using the exponential function exp; just write exp\beta\_i instead of exp^\beta\_i

Section 5 needs to discuss limitations of the presented model(s). E.g. this conceptual modelling approach is not entirely physically based; it builds upon basic principles such as mass preservation and tries to mimic typical macroscopic behaviours of various types of mass movements (e.g., divergence) without modelling the actual internal geomechanical dynamics (e.g. viscous flow etc. as applicable).

Run-up of material on the opposite valley slope doesn't seem to be possible in the GPP model since material is only transferred to lower-elevation neighbouring cells as mentioned in P6L27.

P21L14 "impact of ... immediately obvious" - This is not a result of this study. The authors should avoid claims that are neither based on their findings nor on the cited literature. It may be appropriate to state that the proposed model can be used to assess the potential effectiveness of such forests (provided that the relevant processes are adequately represented by this model and its parameters).

A very brief section outlining implementation details should be included. E.g. which programming language; parallelized implementation? how parallelized, e.g. different Monte Carlo repetitions executed in parallel, . . .?

Is it correct that this model implementation only provides forward modelling capabilities, i.e. modelling possible outcomes based on prescribed model parameters? Or is it also capable of estimating model parameters such as the persistence parameter based on observed runout distributions? Are there any capabilities for validating the model based on observed runout distributions, or does the user have to do this outside the GPP module? I am thinking AUROC estimation based on observed historical events as commonly done in the statistical landslide susceptibility modelling literature.

The presented model and its implementation should be contrasted against other models and software, including commercial products, at least at a general level.

If I understand correctly the present model does not implement scouring (erosion) along the path as implemented by other similar models such as Guthrie et al. (2008) in Landslides.

P21 Section 6 - This shouldn't be a separate section (unless required by journal policy)

Technical comments

P1L7 "practicability" -> applicability or feasibility

P1L8-9 "first ... re-written" seems contradictory; it can't be first if it has been re-written, extended and improved

P1 'large-coverage...' -> 'regional-scale'

P2L26 'disposition modelling ' - susceptibility or slope stability modelling?

P7L21 "like" -> "as"

P7L24 "iterations" -> "repetition"

P7L29 "fixation" – re-word

P9L9 (i.e. first line) and elsewhere: "encoded" - re-word

P20L15 "proven" - re-word

P21L9 "pure" -> "purely"

**2. Author's response to referee 2:**

Geosci. Model Dev. Discuss., doi:10.5194/gmd-2017-5-RC2, 2017 © Author(s) 2017. CC-BY 3.0 License.
"In his manuscript, V. Wichmann presents a GIS-based simulation framework for gravitational processes, i.e. a compilation of model components and their software implementation. It includes various well-known as well as recently developed approaches that conceptually or semi-physically represent displacement processes (but not initiation processes). This simulation framework can be of great interest and value to a broad range of academic, government and corporate users especially since its open-source implementation facilitates access and encourages customization.

While this paper is, in principle, worthy of publication in GMD, in my view the manuscript should still be substantially improved:

- The introductory section currently does not provide a general scientific background and motivation."

**Response:** addressed and added to the introduction

"- The cited references are too narrowly focused on work by Wichmann and Heckmann, including nonrefereed publications. The chosen methods / model components and general modelling approach needs to be situated in the broader context of (physically-based, conceptual and empirical) models of gravitational mass movements."

**Response:** references added; the manuscript has been reworked in order to provide a better distinction between the GPP model components and the implemented modeling approaches – the paper is mainly about the framework of the GPP model, the implemented modeling approaches have been discussed extensively in the cited references; a better classification of the general modeling approach has also been added

"- The Discussion lacks depth. In particular, limitations are not discussed, and comparisons with similar models and software (including commercial products) are missing."

**Response:** I've addressed the limitations and added some comparisons with similar software; I did not add a comparison with commercial products because I don't have access to such software and I feel that such a comparison is out-of-scope of a model description paper

"- The presentation of model structure and components in sections 2 and 3 should be partly re-arranged and re-written as it is often hard to follow (see detailed comments below)."

**Response:** these sections have been partly re-arranged and re-written in order to both provide a better distinction between model components and the implemented modeling approaches and to facilitate the reading/understanding

"I hope that the author will find these general comments as well as the following detailed comments useful in improving their manuscript."

**Detailed comments**

"P1L12 This paper will attract the interest of a broader audience if it starts with a paragraph introducing the motivation for this work and the broader context, e.g. scientific and societal relevance and need for this kind of model and software implementation"

**Response:** two paragraphs were added to the introduction to address these issues

"P1L17 what is 2.5D in this context? perhaps too much detail for an introduction"

Response: the sentence was out of context, removed

"P1L18 Rather than presenting what the author's GPP model is capable of, the author should first provide a brief overview of state-of-the-art modelling approaches for gravitational mass movements (including suitable references to the literature) and then indicate which of these approaches were chosen for / included in GPP and why"

Response: this overview is included in the paragraphs that have been added to the introduction

"P1L20-21 A reference should be included to support this statement. 'simple' may be more appropriate than 'simplistic'"

**Response:** rephrased ("simplifying concepts"); it is now explained in the paragraphs above what is meant by "simplifying"

"P1L23-P2L2 - Some of the Wichmann / Heckmann references are published in less accessible journals and conference proceedings which may overlap in content with some of the peer-reviewed publications by the same authors? A better selection from this set of papers plus additional relevant references to the work of other authors may be more appropriate here."

**Response:** the only non "peer-reviewed" publications are the dissertation of Wichmann (2006), including the most elaborate description of the model components, and the book chapter Wichmann & Becht (2005), which was reviewed by the editors; I don't think the others are published in less

accessible journals; there are no additional references to the work of other authors given as these are, as far as I know, the only studies that are using these modeling approaches for the analysis of sediment cascades and process connectivity.

"P1L23-P2L2 As far as hazard susceptibility modelling is concerned, I noticed that statistical and machine-learning methods (e.g. logistic regression, generalized additive model, support vector machine), which are tremendously popular in this field, aren't mentioned."

**Response:** such models are popular for susceptibility modeling in a broader meaning, i.e. for deriving areas that are affected by mass movements; usually the results describe potential release areas, but not the process path and run-out distance; I've added some information to the introduction

"P2L33 'the plugging of a channel' - check wording; perhaps 'clogged stream channels'?"

**Response:** rephrased to "blocking of a channel by wood and debris"

"P3L5-9 This information is too detailed for a 'general model structure' section. Focus on broad concepts and structures, and explain the general modelling approach. E.g. is the proposed model based on principles of physics or is this a more heuristic GIS-based approach, could it perhaps be referred to as a cellular automata model? The processing steps descript in P3L12-15 appear to suggest a more heuristic approach that certainly ensures mass preservation but is not capable of accommodating the physical (sliding / flowing) behaviour of solid to liquid mixtures of rock, soil, water, snow etc. that may be present in the various types of gravitational mass movements considered here. Such a simple approach may not necessarily be a bad thing, but the methodology should be contrasted appropriately with other possible modelling approaches, in particular physically based ones. In this context it appears to me that the approach presented here is similar to the cellular-automata model proposed by Guthrie et al. (2008) in Landslides in the narrower context of landslide modelling."

**Response:** I've added a more general introduction/overview to this section; the manuscript has been reworked to provide a better distinction between model components and implemented (existing) modeling approaches; a special feature of the GPP model is its modularity, which allows the users to choose the model components to use and which modeling approach in the chosen components should be used; the focus is less on the individual modeling approaches as these have been described and reviewed in many studies; this section is about the general technical layout of the GPP model: which components are implemented (e.g. path finding, velocity calculation, deposition) and how do they interact. The model approaches which can be chosen by the user for each of these components are described in detail in section 3. Information about the principles on which the implemented modeling approaches are based on and how they are categorized, has been added to the introduction. The concepts used in the GPP model are not related to the cellular-automata model proposed by Guthrie et al. (2008). In their approach, landslides are modeled by agents which represent a variable mass of material moving from cell to cell down a slope. The process path is determined by ranking the eight neighbors surrounding a cell, based on their angular difference from the aspect of the current cell. They write: "A continuous probability function is used to describe the spread of the landslide mass to neighbor cells based on a normal distribution centered on the downslope aspect with a standard deviation of 30°". This has only slight similarity to the random walk approach which includes more parameters (slope threshold, exponent of divergence and persistence factor) to control in which

topography and to which extent lateral spreading is simulated. Also the run-out length is determined differently: in their model, if the mass reaches zero, the agent is terminated. The run-out length algorithms used in the GPP model do not consider material deposition.

"P3L5-9 'particle' - In general, I have the feeling that this word may be misleading; at the very least, its meaning in the context of this model should be properly defined. To me the word 'particle' suggests that the model works with some elementary units of mass, e.g. 1 m 3 blocks, that are either passed on or deposited. Does the model really operate on such discrete elementary units, or does it determine amounts of material (e.g. 1.432 m 3, i.e. real values not multiples of discrete units)? - (In P4L8 particles are referred to as "start cells", which adds to the confusion, since a grid cell does not change its location but particles are presumably passed along.)"

**Response:** rephrased; I've added an explanation that the word particle is defined as in physics engines, i.e. as a hypothetical mass point, which is routed downslope

"P4L10-19 - What's the rationale behind these three strategies? What geomechanical process characteristics are they based on?"

**Response:** the processing order of release areas / particles has an influence of the modeling result in case the terrain is modified between two model iterations by sink filling or material deposition. Thus the processing order determines the amount of influence between release areas. Explanation added

"Figs. 1-3 - Are all three figures necessary, or does figure 3 contain all necessary information? Diamond shapes are used for decisions; while "sink" and "stop" may be (nearly) self-explanatory, I am having difficulties understanding why "Material" or "Deposition Model" would involve a yes/no decision. E.g. if the material "stops" (Stop: Yes), then it will be deposited in full - what additional decisions are necessary? Perhaps some re-wording might help, or slightly more detailed labelling of boxes."

**Response:** In principle Fig. 3 does contain all information, but I think it is better to provide three figures which explain the three main model configurations available for the user. Fig. 1 shows the "simplest" configuration, which will be chosen by most users; Fig. 2 and 3 add more and more model components and show how a user can add additional components to the basic setup of the GPP model. Figures have been reworked, including boxes labeling, to make it more obvious that the decisions involve whether a Deposition model is used at all and whether material is left / available for deposition.

"P6, Equation (2): This equation is a bit hard to read at first because of the unusual use of a conditional statement within a set, i.e. two opening curly braces of same size. Also, n\_i should just be i, the cell identifier; n\_i has not been defined. Perhaps to separate equations (2a) and (2b) should be given for N for gamma\_max <= 1 and gamma\_max > 1. A less technical and more direct way of expressing this condition would be to say that all beta\_i are smaller or equal beta\_thres and that at least one adjacent grid cell is steeper than beta\_thres, respectively. Perhaps a brief explanation should be given as to what process or principle this equation is based on. It seems that this explanation is provided in lines P724-30 - why not here, before entering into technical details?"

**Response:** The explanation of the calibration parameters has been moved up and placed before the equations. Additional equations have been added in order to better describe the underlying principle. Equation (2) has been reworked to (2a) and (2b)

"P7L9 "only steep neighbors are allowed" - what is the physical meaning of this? Does this restriction represent a real dynamic process or is this an arbitrary modelling decision?"

**Response:** This conceptual approach is related to the fact that rapid mass movements tend to follow the steepest descent when the topography is steep and that lateral spreading is minimized in those sections of the process path; the higher velocity in such sections is limiting the lateral spreading additionally. In flat topography, in contrast, the velocity is lower, and lateral spreading is usually increased. Modeling approaches developed for hydrological applications do not take this into account. Rephrased.

"P7L11 "which is missing [in] the modeling approaches developed for hydrological processes" - insert "in"; provide reference"

**Response:** inserted and reference added

"In Eq. (4), upper branch (i' \in N), it seems necessary to include the persistence factor p in the denominator, i.e. \sum\_j{p tan\beta\_j} rather than \sum\_j{tan\beta}. In the numerator, placing the factor p before tan\beta\_j would be preferable. With this, the scaling mentioned in L19 seems avoidable as they would already add up to 1."

**Response:** The equation has been completely reworked by introducing a weighting factor which is either 1 or the persistence factor. The scaling is still needed as this scales the probabilities to -accumulated- values, i.e. probability intervals, so that a neighbor can be selected by simply drawing a random number between 0 and 1

"P7L18 "this property (Markov chain)" - the described property, which tries to represent inertia, doesn't seem to be related to the Markov property of stochastic processes."

**Response:** the wording is misleading as I didn't intend a connection to "Markov property" by using the word "property"; although the concept is related to the Markov property because the probability which successor cell is chosen depends on the current state of the cell, I've removed "Markov chain" as it is not necessary and maybe misleading

"P7L29-30 These statements concerning the dynamics properties of various types of mass movements should be supported with suitable references."

Response: references added

"P7L33-38 run-out length calculated with geometric gradient approach - approach not introduced previously. In general, referring to section 3.2.1, how does this fit into the previously described framework that processes mass movements on a cell-by-cell basis while the run-out length approaches look at vectors from initiation points to potential run-out locations?"

**Response:** I've added a reference to Sect. 3.2.2 where the Geometric Gradient approach is described and rephrased. The GPP model includes several different approaches for run-out calculation which the user can choose from. As the run-out length approaches based on angle thresholds are very common, especially as these parameters can be easily derived by field mapping, they are included in the GPP model. For each position along the process path, the angle criterion is checked.

"P8L15 This statement needs to be supported by a reference. The assumptions underlying the estimates presented in this paragraph should be outlined at least briefly."

Response: slightly rephrased; the assumptions are described in detail in the following sections

"P12 Eq. (16) The exponent \beta\_i shouldn't be in superscript when using the exponential function exp; just write exp\beta\_i instead of exp\beta\_i"

**Response:** changed to ex in equations 12 and 16

"Section 5 needs to discuss limitations of the presented model(s). E.g. this conceptual modelling approach is not entirely physically based; it builds upon basic principles such as mass preservation and tries to mimic typical macroscopic behaviours of various types of mass movements (e.g., divergence) without modelling the actual internal geomechanical dynamics (e.g. viscous flow etc. as applicable)."

**Response:** limitations and differences to entirely physically based models have been added to the discussion

"Run-up of material on the opposite valley slope doesn't seem to be possible in the GPP model since material is only transferred to lower-elevation neighbouring cells as mentioned in P6L27."

**Response:** Yes, this is impossible; mentioned as limitation in the discussion

"P21L14 "impact of ... immediately obvious" - This is not a result of this study. The authors should avoid claims that are neither based on their findings nor on the cited literature. It may be appropriate to state that the proposed model can be used to assess the potential effectiveness of such forests (provided that the relevant processes are adequately represented by this model and its parameters)."

**Response:** this is not the result of this study, but has been shown e.g. by Wichmann (2006); I think this is just a misleading wording, rephrased

"A very brief section outlining implementation details should be included. E.g. which programming language; parallelized implementation? how parallelized, e.g. different Monte Carlo repetitions executed in parallel, . . .?"

Response: some details have been added

"Is it correct that this model implementation only provides forward modelling capabilities, i.e. modelling possible outcomes based on prescribed model parameters? Or is it also capable of estimating model parameters such as the persistence parameter based on observed runout distributions? Are there any capabilities for validating the model based on observed runout distributions, or does the user have to do this outside the GPP module? I am thinking AUROC estimation based on observed historical events as commonly done in the statistical landslide susceptibility modelling literature."

**Response:** addressed in the discussion; currently only forward modeling with a focus on process path and run-out length is supported; many features are available in the GIS environment the model is implemented for; model parameter calibration tools would be a great addition

"The presented model and its implementation should be contrasted against other models and software, including commercial products, at least at a general level."

**Response:** the discussion has been reworked, including model limitations and the comparison with other simulation frameworks for gravitational mass movement propagation. I think it is out-of-scope of the paper to discuss individual modeling approaches in detail, as the GPP model is only the framework coupling these established approaches which have been evaluated on their own in various studies

"If I understand correctly the present model does not implement scouring (erosion) along the path as implemented by other similar models such as Guthrie et al. (2008) in Landslides."

Response: yes, currently the GPP model only implements material deposition approaches

"P21 Section 6 - This shouldn't be a separate section (unless required by journal policy)"

Response: journal policy

Technical comments

"P1L7 "practicability" -> applicability or feasibility"

Response: changed to applicability

"P1L8-9 "first ... re-written" seems contradictory; it can't be first if it has been re-written, extended and improved"

**Response:** rephrased

"P1 'large-coverage...' -> 'regional-scale'"

Response: rephrased

"P2L26 'disposition modelling ' - susceptibility or slope stability modelling?"

**Response:** changed to susceptibility

"P7L21 "like" -> "as""

**Response:** changed

```
"P7L24 "iterations" -> "repetition""
```

**Response:** changed

"P7L29 "fixation" - re-word"

Response: rephrased

"P9L9 (i.e. first line) and elsewhere: "encoded" - re-word"

Response: done

"P20L15 "proven" - re-word"

Response: done

"P21L9 "pure" -> "purely""

**Response:** corrected

3. Manuscript with marked changes

**The Gravitational Process Path (GPP) model (v1.0) – a GIS-based simulation framework for gravitational processes**

Volker Wichmann1,2

1alpS, Centre for Climate Change Adaptation, 6020 Innsbruck, Austria
 2Laserdata GmbH, 6020 Innsbruck, Austria
 *Correspondence to:* Volker Wichmann (wichmann@alps-gmbh.com)

**Abstract.** The Gravitational Process Path (GPP) model can be used to simulate the process path and run-out area of gravitational processes based on a digital terrain model (DTM). The tool combines several sub-models conceptual model combines several components (process path, run-out length, sink filling and material deposition) to simulate the movement of a mass point from an initiation site to the deposition area. For each sub-model component several modeling approaches are provided, which makes the tool configurable for different processes like rockfall, debris flows or snow avalanches. The tool can be applied to large-coverage regional scale studies like natural hazard susceptibility mapping on a regional scale but also contains components for scenario based modeling of single events. Both the modeling approaches and precursor implementations of the tool have proven their practicability applicability in numerous studies, including also geomorphological research questions like the delineation of sediment cascades or the study of process connectivity. This is the first completely re-written open source implementation, completely re-written, extended and improved in many ways. The tool has been committed to the main repository of the System for Automated Geoscientific Analyses (SAGA) and thus will be available with every SAGA release.

**1 Introduction**

Rapid mass movements like rockfall, debris flows or snow avalanches, are common features in mountainous regions. Due to population growth and the advancing construction of infrastructure and buildings in such areas, rapid mass movements more and more pose a risk to society and can result in severe damages or even disasters. Besides early warning systems and protection measures for disaster prevention, hazard susceptibility zoning, which identifies potentially endangered areas, is required for risk analysis and the creation of hazard maps (Carrara et al., 1991; Fell et al., 2008; Hu et al., 2016).

While physically based dynamic models can be used for detailed analyses of single events (Takahashi et al., 1992; Iverson, 1997; Pudasaini and Hutter, 2007), regional susceptibility mapping requires modeling approaches with minimal data requirements (Aleotti and Chowdhury, 1999; van Westen and Soeters, 2006; Horton et al., 2013). The input parameters of physically based models are often uncertain, which is why simplified conceptual models are used to estimate potentially endangered areas in regional studies (Mergili et al., 2015). An important part of hazard susceptibility zoning is the description of process paths and run-out distances to determine the objects at risk. This requires to know about potential release areas in order to use these as start points in process path models. Potential process initiation sites can be derived by various methods, including

geomorphological field mapping, the combination of index maps, statistical analyses, deterministic approaches (e.g., factor of safety), probabilistic approaches, or neural networks (Aleotti and Chowdhury, 1999). Originating from the derived starting zones, material, or rather mass points, are then routed over a DTM (digital terrain model). This is done by single or multiple flow direction algorithms, the latter being able to describe lateral spreading away from the slope line (e.g., O'Callaghan and Mark, 1984; Freeman, 1991; Horton et al., 2013). In order to determine the run-out length, often simple break criteria are used like threshold angles based on horizontal and vertical distances (Lied and Bakkehøi, 1980; Hungr and Evans, 1988; Dorren, 2003; Zimmermann et al., 1997). Other approaches, often based on the mass flow model of Voellmy (1955), are using simplified physically based models considering only the centre of mass but not its deformation (Körner, 1976; Perla et al., 1980; Hegg, 1996; Gamma, 2000; Wichmann and Becht, 2005; Horton et al., 2013).

This paper introduces the Gravitational Process Path (GPP) model version 1.0, an attempt to provide a GIS-based modeling framework for the simulation of process path and run-out area of gravitational processes. It combines several modeling approaches in a single tool and simulates the movement of a mass point over a raster DTM (digital terrain model) from an initiation site to the deposition area. It concatenates several sub-models (process path The GPP model is a conceptual model, concatenating components for process path determination, run-out , deposition), each with several modeling approaches, and is calculation, sink filling and material deposition. For each of these components, several well established modeling approaches are implemented and can be chosen by the user. This makes the GPP model configurable for different processes like rockfall, debris flows or avalanches. Working on raster data sets, some of the modeling approaches had to be extended to work in 2.5D.

The GPP model includes stochastic (random walk, Markov chain, Monte Carlo simulation), physically based and empirical Basically, the GPP model simulates the movement of a mass point over a raster DTM from an initiation site to the deposition area. Therefore it includes empirical, stochastic and physically based modeling approaches and provides the option of terrain modification by material deposition during operation. Although some of the approaches are rather simplistic molemented approaches are based on simplifying concepts, realistic results can be achieved with the great advantage of requiring only a few input parameters. This makes it possible to use the tool in large-coverage studies on a regional scale for regional scale studies, but it also includes some components for scenario modeling of single events. Typical applications are natural The approaches implemented in the model components have been successfully used for hazard susceptibility mapping (e.g., Zimmermann et al., 1997; Heinimann et al., 1998; Wichmann and Becht, 2004; Wichmann and Becht, 2005; Mergili et al., 2015; Proske and Bauer, 2016) and geomorphological process studies, e.g. on sediment cascades or process connectivity (e.g., Wichmann, 2006; Wichmann et al., 2009; Haas et al., 2012a; Heckmann et al., 2012; Heckmann and Schwanghart, 2013; Heckmann et al., 2016).

The individual modelingapproaches and model components have proven their applicability to different geomorphological processes and research questions in several studies. The For process path modeling, the GPP model includes the single flow direction path finding approach of O'Callaghan and Mark (1984), also known as the D8 flow direction approach (Jenson and Domingue, 1988), which has been used in various hydrological applications including the derivation of watershed basins and eatchment area. Gamma (1996, 2000) introduced and geomorphological applications. Besides, a random walk approach as introduced in the dfwalk model for debris flow modeling, including a by Gamma (1996, 2000) is implemented. This random walk approach, especially suited for process path delineation of gravitational processes. The random-walk approach, has been

used by various authors for rockfall modeling (e.g., Wichmann and Becht, 2006; Haas et al., 2012b; Proske and Bauer, 2016), debris flow modeling (e.g., Zimmermann et al., 1997; Heinimann et al., 1998; Wichmann and Becht, 2004; Wichmann, 2006; Mergili et al., 2015) and avalanche modeling (e.g., Heckmann, 2006; Schmidtner, 2012).

Run-out distance calculationFor run-out distance calculation, the GPP model includes several approaches based on the energy line principle (e.g., Heim, 1932; Hungr and Evans, 1988)have been used for which have been applied to various processes including rockfall (e.g., Heimimann et al., 1998; Dorren, 2003), debris flows (e.g., Zimmermann et al., 1997) and avalanches (e.g., Körner, 1980). The Besides, the 1-parameter friction model of Scheidegger (1975) is implemented, which has been used for rockfall run-out calculations in several studies (e.g., van Dijke and van Westen, 1990; Meißl, 1998; Dorren and Seijmonsbergen, 2003; Wichmann and Becht, 2005; Wichmann, 2006; Haas et al., 2012b). The avalanche model of Voellmy (1955) and its derivatives, the VSG model (Salm et al., 1990) and the PCM model (Perla et al., 1980), have Finally, the run-out model of Perla et al. (1980), often referred to as PCM model, is included. The PCM model has been applied for avalanche run-out model eling by e.g., Körner (1976), Hegg (1996) and Heckmann (2006). The PCM model It has also been applied to model debris flows (Rickenmann, 1990; Zimmermann et al., 1997; Heinimann et al., 1998; Gamma, 2000; Wichmann, 2006; Mergili et al., 2012; Mergili et al., 2015) and large rock slides (e.g., Körner, 1976).

The GPP model is the first open source implementation based on our previous work previous work of the author, but it is completely reworked and enhanced in various aspects. It is implemented as a tool for the System for Automated Geoscientific Analyses (SAGA, Conrad et al., 2015) and is released as free open-source software (licensed under the GPL). The source code has been committed to the main repository of SAGA hosted at sourceforge.net (https://sourceforge.net/projects/saga-gis/), and binaries are available with every SAGA release.

The paper is structured as follows: Sect. 2 provides an overview of the framework and the model components (process path, run-out, sink filling and deposition). The individual modeling approaches implemented for each component are described in detail in Sect. 3. In Sect. 4 model configurations and application examples for rockfall, debris flow, avalanche and scenario modeling are presented. Finally a discussion and conclusion is provided.

**2 General model structure**

[revised manuscript text omitted]

**3** Modeling approaches**

**4 Model components**

Within the following sections, the currently implemented model approaches of modeling approaches currently implemented for each model component are described in detail. The user can choose which model should be used in each component and combine these selections them to simulate various processes. Typical model configurations are presented in Sect. 4.

**3.1** Process path modeling approaches**

The modeling of process pathways on a raster DTM has been a research topic since many years. The fact that each raster cell has only eight immediate neighbor cells results in problems to reconstruct the correct flow direction over longer distances.

Basically there are two different kinds of methods, single and multiple flow direction algorithms, for which a lot of modeling approacheshave been proposed. A simple (In order to determine the downslope process path of a particle from its initiation site, the GPP model implements two different approaches. One is a single flow direction ) solution has been proposed by O'Callaghan and Mark (1984) algorithm, which selects that neighbor cell as next flow path cell to which the steepest downward slope is observed. Multiple flow direction approaches (e.g., Freeman, 1991) usually distribute the accumulated water or material among all neighbor cells to which a downward slope is recognized. But most of these approaches have been developed for hydrological applications and are only of limited use in order to model gravitational processes: the amount of water is usually distributed in more or less the same proportions to the neighbors, irrespective of the local slope conditions. Therefore Gamma (1996, 2000) introduced the *mfdf* approach (multiple flow directions for debris flows) which is The other, based on a random walk, is a multiple flow direction approach sensitive to the local slope conditions.

**3.1.1 Maximum slope**

This approach, as proposed by O'Callaghan and Mark (1984), is implemented mainly for convenience in order to provide a simple means to detect the process path along the gradient of gravity. A particle follows the steepest descent of the slope:

$$n = max\{(z - z_i)/d_i\}\tag{1}$$

where *n* is the neighbor of steepest descent, *z* is the elevation of the currently processed cell,  $z_i$  is the elevation of neighbor cell *i*, and  $d_i$  is the horizontal distance to neighbor cell *i*.

The model result is thus deterministic, with the exception of its behavior (as implemented in the GPP model) when two or more neighbor cells show the same steepest descent or when a flat area is reached. In the first case, one of the neighbors cells is chosen by random. On flat areas a set of potential neighbor cells is determined which is made up of all neighbors with the same elevation as the current cell which have not been traversed yet in the current model iteration. From this set, a process path cell is chosen by random. Together with the possibility that the terrain could be modified. This introduces a probabilistic component. Further, the terrain could have been modified between two model iterations by sink filling or material depositionbetween two model iterations, this introduces a probabilistic component.

The *Maximum Slope* model approach has no special parameters besides those controlling the mode of operation of the GPP model main loop, like the number of model operations repetitions or the processing order. The pseudo-random number generator, used to choose a neighbor cell by random under the pre-described conditions, can be initialized either with the current time or a fixed seed value. The latter will always produce the same succession of values for a given seed value and will thus give the same results for consecutive tool runs.

**3.1.2 Random walk**

With this approach, the process path is modeled by a variant of the dfwalk model of Gamma (2000). The model can be adjusted to as proposed by Gamma (1996, 2000). Besides the parameters controlling the Monte Carlo simulation like the number of

repetitions, the *Random Walk* approach has three parameters to calibrate the model in order to mimic the behavior of different geomorphological processesby three calibration parameters.: (i) a slope threshold controls below which terrain slope divergent flow is allowed; (ii) this is accompanied by an exponent for divergent flow: below the slope threshold, the parameter controls the degree of divergence; (iii) finally, a persistence factor can be used to preserve the direction of movement by weighting the current flow direction in order to account for inertia, which can be observed for debris flows or wet snow avalanches (Nohguchi, 1989; Takahashi et al., 1992). Rockfall may be modeled with (almost) no persistence and a higher degree of divergence.

For the currently processed grid cell, a set N of potential flow path cells is determined from all immediate neighbor cells in a 3 by 3 window<del>which have a</del>, which have an equal or lower elevation than the central cell. There are two parameters available – a slope threshold and a parameter controlling divergent flow – to further reduce this set. Possible flow path cells are determined by the *mfdf* criterion. This is done in several steps. First of all, for each neighbor cell *i* a slope value  $\gamma_i$ , based on the slope threshold  $\beta_{threes}$ , is calculated (Gamma, 2000; Wichmann and Becht, 2005):

$$\mathbf{N} = \left\{ n_i \mid \left\{ \begin{array}{ll} \gamma_i \ge (\gamma_{max})^a & \text{if } 0 < \gamma_{max} \le 1\\ \gamma_i = \gamma_{max} & \text{if } \gamma_{max} > 1 \end{array} \right., \quad i \in \{1, 2, \dots 8\}, \quad a \ge 1 \right\}$$

and-

$$\gamma_i = \frac{\tan \beta_i}{\tan \beta_{thres}}, \qquad \beta_i \ge 0, \qquad i \in \{1, 2, \dots 8\}$$

$$\tag{2}$$

where  $\gamma_{max}$  is the  $max\{\gamma_i\}$ ,  $\beta_i$  is the slope to neighbor cell i,  $\beta_{thres}$  is a slope threshold and. The maximum value  $\gamma_{max} = max\{\gamma_i\}$  is a measure on how close the slope to the steepest neighbor is to the slope threshold. In case  $\gamma_{max} > 1$  the set **N** of potential flow path cells is only made up of the steepest neighbor. Otherwise, the *mfdf* (multiple flow directions for debris flows; Gamma, 2000) criterion is used to decide which neighbor cells are additionally included in **N**:

$$\gamma_i \ge (\gamma_{max})^a \quad (0 < \gamma_{max} \le 1, \quad a \ge 1) \tag{3}$$

where a is an the exponent to control the amount of divergent flow. If  $\gamma_i$  is greater than or equal to the *mfdf* criterion, then the neighbor i is included in N. Thus, the set N is given by either:

$$\mathbf{N} = \{i \mid \gamma_i \ge (\gamma_{max})^a \qquad \qquad \text{if } 0 < \gamma_{max} \le 1, \qquad \qquad i \in \{1, 2, \dots 8\}, \qquad \qquad a \ge 1 \qquad (4a)$$

 $\mathbf{N} = \{i \mid \gamma_i = \gamma_{max} \qquad \qquad \text{if } \gamma_{max} > 1, \qquad \qquad i \in \{1, 2, \dots 8\}$ (4b)

The slope threshold makes it possible to adjust the model to different relieftopography: in steep sections of the process path, where the terrain slope is near the threshold, only steep neighbors are allowed in addition to the steepest descent. In flat

sections, almost all lower neighbor cells are potential flow path cells and the tendency for divergent flow is increased. This The degree of divergent flow below the slope threshold can be controlled by the exponent of divergent flow. This sensitivity to the terrain conditions is an important property which is missing in the modeling approaches developed for hydrological processes. The degree of divergent flow can be controlled by parameter *a*, which distribute the flow proportionally to the slope to all lower neighbors irrespective of the local topography (Gamma, 2000).

From the final set NFinally, a cell is picked by random from the set N. The probability for each cell  $p_i$  probi is given by

$$\underline{pprob}_{i} = \underline{i, j \in \mathbf{N}, p1} \underbrace{\frac{f_{i} \cdot \tan \beta_{i}}{\sum_{j} f_{j} \cdot \tan \beta_{j}}}_{\sum_{j} f_{j} \cdot \tan \beta_{j}}$$
(5)

where i' denotes the previous flow direction and p is a persistence factor(which is also contained in the computation of the sum if  $i' \in \mathbb{N}$ )*i* describes the currently processed neighbor cell, *j* depicts all neighbor cells in set  $\mathbb{N}$ , and *f* is a weighting factor. In case the flow direction to neighbor *i* equals the previous flow direction, *f* equals the persistence factor *p* (with  $p \ge 1$ ), otherwise f = 1. A tendency to move towards the steepest descent is always achieved given as the transition probabilities are weighted by slope. In case the previous flow direction *i'* is contained in the set  $\mathbb{N}$ , the persistence factor is used to give this direction higher weight and thus a higher probability to get. The persistence factor can be used to weight the current flow direction, which results in a higher probability that the neighbor in this direction gets selected. This property (Markov Chain) can be used to reduce abrupt changes in flow direction. Finally the transition probabilities are scaled to accumulated values between 0 and 1, and the pseudo-random generator is used to select one flow path cell from the set.

In the GPP model, the approach is extended to also handle flat areas. This is done like as described for the *Maximum Slope* approach with the same restriction that a potential successor cell must not have been traversed yet in the current model iteration in order to prevent endless loops.

Besides the parameters controlling the Monte Carlo simulation like the number of iterations, the *Random Walk* approach has three parameters to calibrate the model in order to mimic the behavior of different geomorphological processes. The *mfdf* criterion (Eq. (4)) controls below which terrain slope divergent flow is allowed. Multiple neighbors are only allowed in case the steepest local slope is lower than the slope threshold. This is accompanied by the exponent for divergent flow: below the slope threshold, the parameter controls the degree of divergence. Finally, the persistence factor can be used to achieve a greater fixation in the direction of movement (accounting for inertia) as may be the case for debris flows or wet snow avalanches. Roekfall may be modeled with (almost) no persistence and a higher degree of divergence.

The result of several model iterations is a raster data set with encoded storing the transition frequencies, i.e. how many times a grid cell has been traversed. Figure 4 shows the effect of different parameter settings for the three calibration parameters slope threshold, exponent for divergent flow and persistence factor  $\cdot$ . Here, (the run-out length was calculated with the *Geometric Gradient* approach using an angle of 26.5°, see Sect. 3.2.2). The number of model iterations is set to 1000 in the examples (a) to (j). In Fig. 4 (a) to (e) the slope threshold (40°) and the persistence factor (1.0) are fixed, while the exponent for divergent flow is increased in several steps (1.0, 1.1, 1.2, 1.5, and 2.0). It is obvious that the extent of the process area increases significantly because of the higher degree of lateral spreading.

---

## Referee Report (RR1)

**Technical corrections:**

Page 1, line 19: „needs" instead of „requires" in order to avoid the expression" requires …. with requirements"

Page 1, line 23: "this requires to know about",

Page 2, line 3: "can be routed" instead of "are routed", "can be done" instead of "is done", as there are always also other possibilities: some models don't use mass points but irregularly formed blocks (e.g. RAMMS), some models use vector paths on TINs (e.g. described by Hegg 1997, Geographica Bernensia 52)

Page 2, line 27: "It is" instead of "this random walk approach," in order to avoid repetitions.

Page 4, line 2: What does "the particle is destroyed" mean? Maybe better: "deleted"

Page 5, line 3: it think "has not stopped" is correct instead of "did not stop"

Page 7, line 11: an explanation of the random walk would be helpful, such as "…, a stochastic way of path finding, enabling in case of repeated use to model the lateral spread of process areas."

Page 9, line 7: "in the GPP several approaches are implemented" instead of "the GPP implements several approaches" as the implementation is not done by GPP but by the programmer.

Page 10: As the suggested renumbering of the subchapters is not possible due to formatting rules, I would suggest to skip the numbers and use letters: a) Geometric gradient, b) Fahrboeschung, c) Shadow angle. The subheading "energy line approaches" should be one step above in the chapter hierarchy. I find it confusing when it is on the same level.

Page 13, line 17: "in the GPP equation (17) is implemented" instead of "the GPP implements equation (17)".

Page 13, line 25: implement (use passive instead of active) line 25: analogous to the comments above.

Page 15, line 17 f.: "This section provides a brief summary on the GPP model parameters, input, and output data sets." leads to the expectation that a longer passage would follow but in reality in this section only references to tables in the supplement are given. Thus I would change the sentence: "A brief summary on the GPP model parameters, input, and output data sets is given in the supplement:"

Page 21, line 15: "build on" instead of "build upon"

---

## Editor Decision (ED1)

X

x

July 21, 2017

**Contents**

Thus, the set N is given by (Gamma, 2000):

$$N = \{i | \gamma_i \geq (\gamma_{max})^a \text{ for } i \in \{1, 2, 3 \ldots 8\}\} \tag{1}$$

if $\gamma_{max} < 1$ and

$$N = \{i | \gamma_i = \gamma_{max} \text{ for } i \in \{1, 2, 3 \ldots 8\}\} \tag{2}$$

if $\gamma_{max} \geq 1$.

---

## Author Response (AR2)

**Author's response**

"The Gravitational Process Path (GPP) model (v1.0) – a GIS-based simulation framework for gravitational processes" by Volker Wichmann

Dear Getraud Meißl,

thank you very much for your remarks and technical corrections. I've incorporated these into the manuscript.

This response is structured as follows: the first section (as required by journal policy) includes your comments. In section two I have added my responses to the comments. Finally a document which highlights the actual changes made to the document is attached.

Thanks again and best regards,

Volker Wichmann

**1) Comments from referee 1**

Technical corrections:

Page 1, line 19: „needs" instead of „requires" in order to avoid the expression" requires .... with requirements"

Page 1, line 23: "this requires to know about",

Page 2, line 3: "can be routed" instead of "are routed", "can be done" instead of "is done", as there are always also other possibilities: some models don't use mass points but irregularly formed blocks (e.g. RAMMS), some models use vector paths on TINs (e.g. described by Hegg 1997, Geographica Bernensia 52)

Page 2, line 27: "It is" instead of "this random walk approach," in order to avoid repetitions.

Page 4, line 2: What does "the particle is destroyed" mean? Maybe better: "deleted"

Page 5, line 3: it think "has not stopped" is correct instead of "did not stop"

Page 7, line 11: an explanation of the random walk would be helpful, such as "..., a stochastic way of path finding, enabling in case of repeated use to model the lateral spread of process areas."

Page 9, line 7: "in the GPP several approaches are implemented" instead of "the GPP implements several approaches" as the implementation is not done by GPP but by the programmer.

Page 10: As the suggested renumbering of the subchapters is not possible due to formatting rules, I would suggest to skip the numbers and use letters: a) Geometric gradient, b) Fahrboeschung, c) Shadow angle. The subheading "energy line approaches" should be one step above in the chapter hierarchy. I find it confusing when it is on the same level.

Page 13, line 17: "in the GPP equation (17) is implemented" instead of "the GPP implements equation (17)".

Page 13, line 25: implement (use passive instead of active) line 25: analogous to the comments above.

Page 15, line 17 f.: "This section provides a brief summary on the GPP model parameters, input, and output data sets." leads to the expectation that a longer passage would follow but in reality in this section only references to tables in the supplement are given. Thus I would change the sentence: "A brief summary on the GPP model parameters, input, and output data sets is given in the supplement:"

Page 21, line 15: "build on" instead of "build upon"

**2. Author's response to referee 1:**

Wichmann, V. (2017): The Gravitational Process Path (GPP) model (v1.0) – a GIS-based simulation framework for gravitational processes. GMD

Technical corrections:

Page 1, line 19: „needs" instead of „requires" in order to avoid the expression" requires .... with requirements"

**Response:** changed

Page 1, line 23: "this requires to know about",

**Response:** left unchanged as it is unclear to me what should be corrected

Page 2, line 3: "can be routed" instead of "are routed", "can be done" instead of "is done", as there are always also other possibilities: some models don't use mass points but irregularly formed blocks (e.g. RAMMS), some models use vector paths on TINs (e.g. described by Hegg 1997, Geographica Bernensia 52)

**Response:** changed

Page 2, line 27: "It is" instead of "this random walk approach," in order to avoid repetitions.

**Response:** changed

Page 4, line 2: What does "the particle is destroyed" mean? Maybe better: "deleted"

**Response:** "destroyed" is C++ nomenclature and basically means that an object is deleted, changed to "deleted"

Page 5, line 3: it think "has not stopped" is correct instead of "did not stop"

**Response:** changed

Page 7, line 11: an explanation of the random walk would be helpful, such as "..., a stochastic way of path finding, enabling in case of repeated use to model the lateral spread of process areas."

**Response:** I've added the following sentence: "It uses a stochastic way of path finding, which makes it possible to model the lateral spreading of a process by calculating several iterations from the same start position."

Page 9, line 7: "in the GPP several approaches are implemented" instead of "the GPP implements several approaches" as the implementation is not done by GPP but by the programmer.

**Response:** rephrased

Page 10: As the suggested renumbering of the subchapters is not possible due to formatting rules, I would suggest to skip the numbers and use letters: a) Geometric gradient, b) Fahrboeschung, c) Shadow angle. The subheading "energy line approaches" should be one step above in the chapter hierarchy. I find it confusing when it is on the same level.

**Response:** changed to (a), (b), and (c)

Page 13, line 17: "in the GPP equation (17) is implemented" instead of "the GPP implements equation (17)".

**Response:** changed

Page 13, line 25: implement (use passive instead of active) line 25: analogous to the comments above.

**Response:** changed

Page 15, line 17 f.: "This section provides a brief summary on the GPP model parameters, input, and output data sets." leads to the expectation that a longer passage would follow but in reality in this section only references to tables in the supplement are given. Thus I would change the sentence: "A brief summary on the GPP model parameters, input, and output data sets is given in the supplement:"

**Response:** rephrased

Page 21, line 15: "build on" instead of "build upon"

**Response:** changed

**3. Manuscript with marked changes**

**Author's response**

Geosci. Model Dev. Discuss.,
doi:10.5194/gmd-2017-5-RC2, 2017

"The Gravitational Process Path (GPP) model (v1.0) – a GIS-based simulation framework for gravitational processes" by Volker Wichmann

Dear anonymous reviewer,

thank you very much for your remarks and technical corrections. I've incorporated these into the manuscript.

This response is structured as follows: the first section (as required by journal policy) includes your comments. In section two I have added my responses to the comments. Finally a document which highlights the actual changes made to the document is attached.

Thanks again and best regards,

Volker Wichmann

**1) Comments from referee 2**

The author has presented a substantially improved version of his manuscript on gravitational process path (GPP) modelling. The revised manuscript is better motivated and organized, its language has been much improved, and relevant references have been added.

Nevertheless I would still like to see some additional changes after which the manuscript should be publishable, at least as far as I'm concerned.

- Equations (4a,b) in combination with (3) still needs to be improved to make sense. N is not a meaningful definition of a set since the curly braces won't close. Eq. (4a) seems to be for a>=1 and Eq. (4b) otherwise, but it seems that different ranges of \gamma_max also apply to these equations. So how would the set be determined for a<1 and \gamma_max < 1, for example? HAsn't this been published somewhere (Gamma, 2000?) where it can be looked up? Or could the author explain the set verbally, as in "For a\ge1 and 0<\gamma_max<1, N is the set of all neighbours i\in\{1,…,8\} with \gamma_i\ge\gamma_max^a. For a<1, …"

- Self-citations can still be reduced. There are instanced where references are rather 'optional' (where the text says 'e.g. …' as on P2) and several Wichmann / Heckmann references are listed along with a small number of references by other authors. While I am not arguing that Wichmann / Heckmann studies aren't important, many readers will get the impression that these references are over-represented.

- P22 "physically based numerical simulation models are available" – add references

- P23 "with great cautiousness" -> "carefully"

**2. Author's response to referee 2:**

The author has presented a substantially improved version of his manuscript on gravitational process path (GPP) modelling. The revised manuscript is better motivated and organized, its language has been much improved, and relevant references have been added.

Nevertheless I would still like to see some additional changes after which the manuscript should be publishable, at least as far as I'm concerned.

- Equations (4a,b) in combination with (3) still needs to be improved to make sense. N is not a meaningful definition of a set since the curly braces won't close. Eq. (4a) seems to be for a>=1 and Eq. (4b) otherwise, but it seems that different ranges of \gamma_max also apply to these equations. So how would the set be determined for a<1 and \gamma_max < 1, for example? HAsn't this been published somewhere (Gamma, 2000?) where it can be looked up? Or could the author explain the set verbally, as in "For a\ge1 and 0<\gamma_max<1, N is the set of all neighbours i\in\{1,…,8\} with \gamma_i\ge\gamma_max^a. For a<1, …"

**Response:** I've replaced 4a/b with the original equation as published by Gamma (2000). This is more or less the same equation as committed with the first version of the manuscript, but as I just noticed, there was a problem in formula typesetting in my first version, sorry for that! I've also added a statement that a is always >= 1 in the text to make this clearer. So I think the set should be completely defined now.

- Self-citations can still be reduced. There are instanced where references are rather 'optional' (where the text says 'e.g. …' as on P2) and several Wichmann / Heckmann references are listed along with a small number of references by other authors. While I am not arguing that Wichmann / Heckmann studies aren't important, many readers will get the impression that these references are over-represented.

**Response:** I've reduced the references further

- P22 "physically based numerical simulation models are available" – add references

**Response:** added

- P23 "with great cautiousness" -> "carefully"

**Response:** changed

**3. Manuscript with marked changes**

[revised manuscript text omitted]

---

## Author Response (AR3)

**Author's response**

"The Gravitational Process Path (GPP) model (v1.0) – a GIS-based simulation framework for gravitational processes" by Volker Wichmann

Hello Lutz,

thank you for your suggestion, that makes sense.

This response is structured as follows: the first section includes the editor comment. In section two I have added my response to the comment. Finally a document wichmann_gpp_model_diff.pdf is attached which highlights the actual changes made to the document.

Thanks again and best regards,

Volker Wichmann

**1) Comments from the editor**

Topical Editor Decision: Publish subject to minor revisions (Editor review) (21 Jul 2017) by Lutz Gross
Comments to the Author (pdf): gmd-2017-5-comments-to-author.pdf
Comments to the Author:
Volker

Thanks for uploading a new manuscript.
Sorry for being a bit pedantic but your older version of equations (4) made more sense to me.
I think what you need do is to move the condition on gamma_max>0 out of the set definition, see attached file to get the idea. It is also my understanding that in your notation i is the index of neighboring node not n_i

Thanks.
Lutz

**2) Author's response to the editor**

Thanks for uploading a new manuscript.
Sorry for being a bit pedantic but your older version of equations (4) made more sense to me.
I think what you need do is to move the condition on gamma_max>0 out of the set definition, see attached file to get the idea. It is also my understanding that in your notation i is the index of neighboring node not n_i

**Response:** Thanks for the suggestion, I've reformatted the formula accordingly.

**3) Manuscript changes**

[revised manuscript text omitted]